# Bardet–Biedl syndrome 3 protein promotes ciliary exit of the signaling protein phospholipase D via the BBSome

Yan-Xia Liu[1†], Bin Xue[1†], Wei-Yue Sun[1], Jenna L Wingfield[2], Jun Sun[1], Mingfu Wu[3], Karl F Lechtreck[2], Zhenlong Wu[4], Zhen-Chuan Fan[1]*

[1]State Key Laboratory of Food Nutrition and Safety, Institute of Health Biotechnology, Tianjin University of Science and Technology, Tianjin, China; [2]Department of Cellular Biology, University of Georgia, Athens, United States; [3]Department of Molecular and Cellular Physiology, Albany Medical College, Albany, United States; [4]State Key Laboratory of Animal Nutrition, China Agricultural University, Beijing, China

**Abstract** Certain ciliary signaling proteins couple with the BBSome, a conserved complex of Bardet–Biedl syndrome (BBS) proteins, to load onto retrograde intraflagellar transport (IFT) trains for their removal out of cilia in *Chlamydomonas reinhardtii*. Here, we show that loss of the Arf-like 6 (ARL6) GTPase BBS3 causes the signaling protein phospholipase D (PLD) to accumulate in cilia. Upon targeting to the basal body, BBSomes enter and cycle through cilia via IFT, while BBS3 in a GTP-bound state separates from BBSomes, associates with the membrane, and translocates from the basal body to cilia by diffusion. Upon arriving at the ciliary tip, GTP-bound BBS3 binds and recruits BBSomes to the ciliary membrane for interacting with PLD, thus making the PLD-laden BBSomes available to load onto retrograde IFT trains for ciliary exit. Therefore, BBS3 promotes PLD exit from cilia via the BBSome, providing a regulatory mechanism for ciliary signaling protein removal out of cilia.

*For correspondence:
fanzhen@tust.edu.cn

[†]These authors contributed equally to this work

Competing interests: The authors declare that no competing interests exist.

## Introduction

The cilium consists of a microtubule-based axoneme surrounded by a specialized ciliary membrane and is assembled relying on motor-driven intraflagellar transport (IFT) trains, which consist of repeating units of the multiprotein complexes IFT-A and IFT-B (composed of IFT-B1 and -B2 subcomplexes) and traffic along the axoneme (*Cole et al., 1998*; *Follit et al., 2009*; *Kozminski et al., 1993*; *Lucker et al., 2005*; *Nachury, 2018*; *Ou et al., 2007*; *Pigino et al., 2009*; *Taschner et al., 2014*; *Taschner et al., 2016*). As a signaling hub, the cilium senses and transmits extracellular stimuli inside the cell through ciliary signaling proteins, including G protein-coupled receptors (GPCRs), ion channels, and others such as phospholipase D (PLD) and AMP-activated protein kinase (*Berbari et al., 2008*; *Datta et al., 2015*; *Domire et al., 2011*; *Liu and Lechtreck, 2018*; *Siljee et al., 2018*; *Valentine et al., 2012*; *Zhang et al., 2012*). These membrane proteins cycle between cell membrane and ciliary membrane via IFT (*Kozminski et al., 1993*; *Nachury, 2018*; *Pigino et al., 2009*). During this process, the octameric BBSome, composed of eight Bardet–Biedl syndrome (BBS) proteins (BBS1/2/4/5/7/8/9/18), acts as an IFT adaptor to link specific ciliary signaling proteins to IFT trains for their transport in and/or out of the cilium (*Bhogaraju et al., 2013*; *Eguether et al., 2014*; *Lechtreck et al., 2009*; *Liew et al., 2014*; *Liu and Lechtreck, 2018*; *Nachury, 2018*; *Nachury et al., 2007*; *Ruat et al., 2012*; *Su et al., 2014*; *Zhang et al., 2012*). Therefore, defects in assembly and composition of the BBSome or factors required for mediating ciliary cycling of the BBSome lead to loss and/or abnormal accumulation of signaling proteins in the ciliary membrane

(*Chiang et al., 2004*; *Lechtreck et al., 2009*; *Loktev et al., 2008*; *Nachury et al., 2007*; *Scheidecker et al., 2014*; *Zhang et al., 2011*). This eventually causes improper ciliary signaling and results in BBS, a human inherited ciliopathic disorder characterized by multiorgan dysfunction such as retinal dystrophy, male infertility, polydactyly, early-onset obesity, and renal failure in some cases (*Fliegauf et al., 2007*).

The BBSome performs IFT-dependent ciliary cycling in four continuous steps: coupling with IFT at the ciliary base, entry and anterograde traffic (from the base to the tip), remodeling and turnaround at the ciliary tip, and retrograde traffic (from the tip to the base) and ciliary exit (*Eguether et al., 2014*; *Iomini et al., 2001*; *Keady et al., 2012*; *Lechtreck et al., 2013*; *Liew et al., 2014*; *Pedersen et al., 2006*; *Pedersen et al., 2005*; *Wei et al., 2012*). During this process, BBSome cargoes, such as somatostatin receptor 3 (Ssr3) and melanin-concentrating hormone receptor 1 (Mchr1), couple with the BBSome at the ciliary base for ciliary entry (*Berbari et al., 2008*; *Jin et al., 2010*). In contrast, others like dopamine receptor 1 (D1), GPR161, and PLD load onto the BBSome at the ciliary tip for ciliary exit (*Domire et al., 2011*; *Liew et al., 2014*; *Liu and Lechtreck, 2018*; *Ye et al., 2018*). It was shown that the BBSome is recruited from the cell body to the ciliary base as the major effector of the Arf-like 6 (ARL6) GTPase BBS3 (*Jin et al., 2010*; *Xue et al., 2020*). Such a scenario could allow cells to control the amount of BBSomes available at the basal body and in turn regulate the presence and amount of signaling protein cargoes in the ciliary membrane (*Jin et al., 2010*; *Xue et al., 2020*). When in its GTP-bound state, BBS3 enters cilia (*Liew et al., 2014*; *Xue et al., 2020*) and undergoes GTPase cycling (*Liew et al., 2014*). Upon finishing a GTPase cycle with the aid of the Rab-like 4 (RABL4) GTPase IFT27 as a BBS3-specific guanine nucleotide exchange factor (GEF), GTP-loaded BBS3 mimics its role in the cell body to bind the cargo-laden BBSomes at the ciliary tip and loads cargoes onto retrograde IFT trains for ciliary exit, thus regulating the cargo content in the ciliary membrane (*Liew et al., 2014*). Currently, the precise molecular activity by which BBS3 maintains the ciliary dynamics of signaling protein cargoes through the BBSome remains to be determined, while both *in vitro* biochemical analysis and structural studies have implicated that membrane association of BBS3 in its active GTP-bound state is a prerequisite for signaling protein cargoes to couple with BBSomes (*Jin et al., 2010*; *Liew et al., 2014*; *Loktev et al., 2008*; *Mourão et al., 2014*; *Nachury et al., 2007*).

Our current limited understanding of how BBS3 and the BBSome interact for regulating signaling protein content in the ciliary membrane was derived mainly from biochemical assays performed on whole-cell extracts but not ciliary extracts of mammalian cells (*Jin et al., 2010*; *Liew et al., 2014*). *Chlamydomonas reinhardtii* has a clear advantage over other ciliated model organisms as its cilia can be easily isolated for biochemical analysis. Hence, we explored how BBS3 and the BBSome cross-talk in *C. reinhardtii* for targeting to the basal body, for entering cilia from the basal body, and for maintaining their dynamics in cilia and how they coordinate to mediate ciliary exit of PLD in *C. reinhardtii* (*Liu and Lechtreck, 2018*). By performing functional, *in vivo* biochemical, and single-particle *in vivo* imaging assays, we showed that the BBSome depends on BBS3 for targeting to the basal body but not *vice versa*. Upon targeting to the basal body, GTP-bound BBS3 separates from the BBSome and they enter cilia by diffusion (BBS3) and via IFT (the BBSome). Upon reaching the ciliary tip, BBS3 mediates the sorting of PLD out of cilia through promoting its coupling with the BBSome, thus filling a gap in our understanding of how BBS3 regulates ciliary removal of signaling proteins through the BBSome in *C. reinhardtii*.

## Results

### BBS3 is not required for the BBSome to enter cilia from the basal body

It was previously reported that the BBSome relies on BBS3 for entering cilia in human cells (*Jin et al., 2010*). Our previous study showed that *Chlamydomonas* BBS3 is actually required for recruiting the BBSome to the basal body, thus making the BBSome available for loading onto anterograde IFT trains for ciliary entry (*Xue et al., 2020*). This finding raises the question of whether the BBSome, upon targeting to the basal body, requires BBS3 for entering cilia. To answer this question, we examined the so-called BBS3 CLiP mutant (LMJ.RY0402.149010) that we named *clip1*. The *clip1* strain contains a 1923 bp paromomycin gene insertion in 3' distal region of the fifth intron of the BBS3 gene (*Figure 1—figure supplement 1A–C*). This insertion does not cause changes in BBS3

cDNA sequence (*Figure 1—figure supplement 1D, E*), and the endogenous BBS3 and the BBSome subunits BBS1 and BBS5 were maintained at wild-type levels (*Figure 1A*), demonstrating that the *clip1* strain is not a real BBS3-null mutant. Strikingly, BBS3 was not present in cilia of the *clip1* strain, while BBS1 and BBS5 were maintained at wild-type levels in cilia, a result showing that the BBSome does not require BBS3 for entering cilia (*Figure 1B*). To verify that BBS3 is not able to enter cilia in *clip1* cells, we expressed BBS3 attached at its C-terminus to a yellow fluorescent protein (YFP) (BBS3::YFP) in wild-type CC-5325 and *clip1* cells (resulting strains BBS3::YFP[5325] and BBS3::YFP[clip1]) (*Figure 1C*). The transgenic BBS3::YFP, similar to the endogenous BBS3, was present in cilia of BBS3::YFP[5325] cells but not of BBS3::YFP[clip1] cells (*Figure 1D, E*). As expected, BBS1 and BBS5 were maintained at wild-type levels in whole-cell samples and cilia of both strains (*Figure 1C, D*). In addition, BBS5 and BBS3::YFP were concentrated and colocalized at the basal bodies in both BBS3::YFP[5325] and BBS3::YFP[clip1] cells (*Figure 1E*), and BBS3::YFP immunoprecipitated BBS1 and BBS5 in the cell body extracts of BBS3::YFP[clip1] cells (*Figure 1F*), suggesting that BBS3 retains its ability to bind and recruit the BBSome to the basal body in *clip1* cells (*Xue et al., 2020*). Thus, it remains to

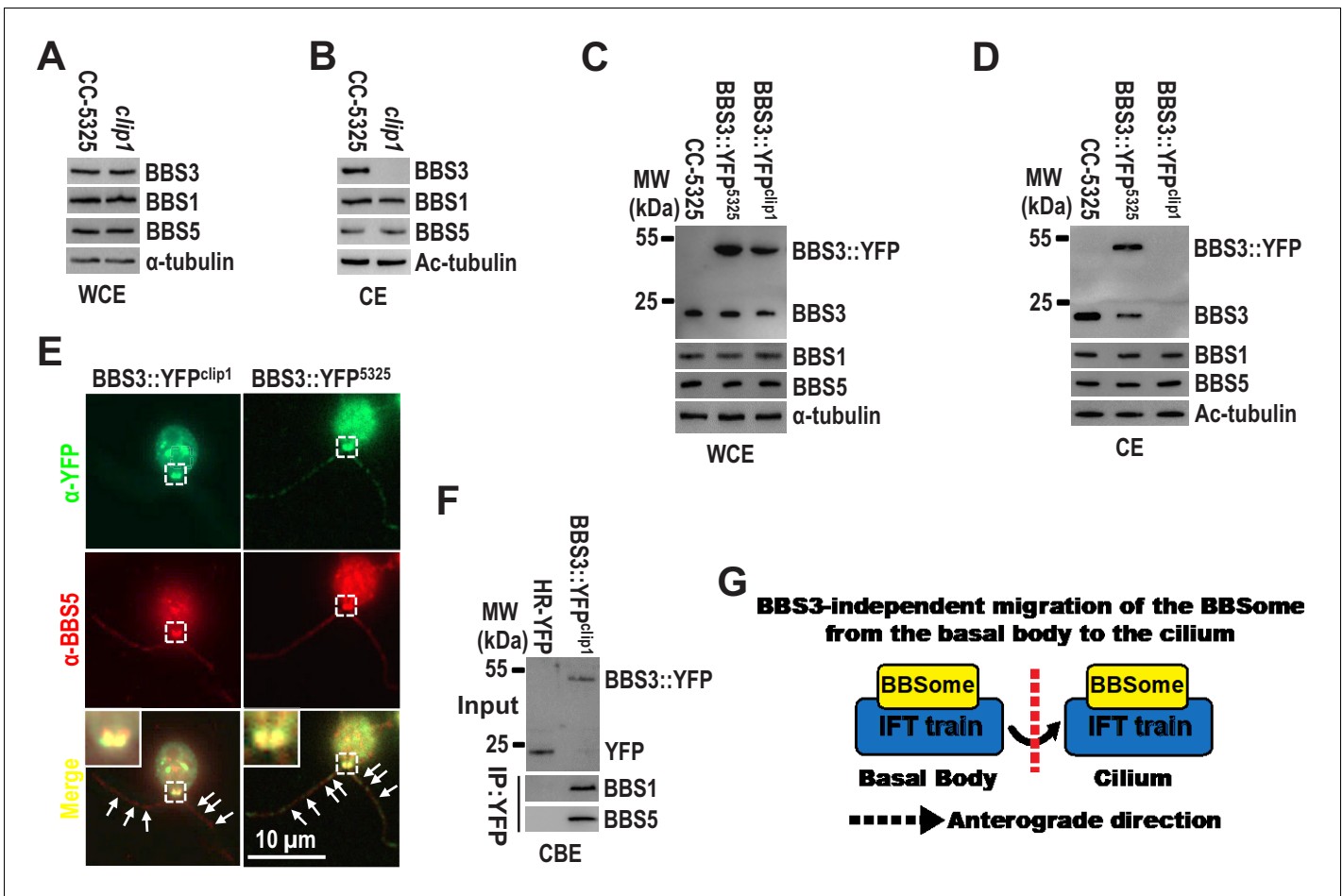

**Figure 1.** BBS3 is not required for the BBSome to enter cilia from the basal body. (**A and B**) Immunoblots of whole-cell extracts (WCE) (**A**) and ciliary extracts (CE) (**B**) of CC-5325 and *clip1* cells probed for BBS3, BBS1, and BBS5. (**C and D**) Immunoblots of WCE (**C**) and CE (**D**) of CC-5325, BBS3::YFP[5325], and BBS3::YFP[clip1] cells probed with α-BBS3, α-BBS1, and α-BBS5. MW: molecular weight. (**E**) BBS3::YFP[5325] and BBS3::YFP[clip1] cells stained with α-YFP (green) and α-BBS5 (red). Inset shows the basal bodies. White arrows show ciliary staining. Scale bar 10 μm. (**F**) Immunoblots of α-YFP captured proteins from cell body extracts (CBE) of HR-YFP (YFP-expressing CC-125 cells) and BBS3::YFP[clip1] cells probed with α-BBS1 and α-BBS5. Input was quantified with α-YFP by immunoblotting. (**G**) Schematic presentation showing that the BBSome migrates from the basal body into cilia via IFT in a BBS3-independent manner. For immunoblotting, α-tubulin and acetylated-tubulin (Ac-tubulin) were used to adjust the loading of WCE and CE, respectively.

The online version of this article includes the following figure supplement(s) for figure 1:

**Figure supplement 1.** Characterization of the CLiP mutant *clip1* (LMJ.RY0402.149010) at genomic and cDNA levels.

be investigated why BBS3 concentrates at the basal bodies but is prevented from entering cilia in *clip1* cells, while our data clearly show that *Chlamydomonas* BBS3 is not required for entry of the BBSome into cilia from the basal bodies (*Figure 1G*). This notion was confirmed when BBS3 knock-down and rescue studies were performed (Figure 7D, E).

## BBS3 enters cilia without relying on the BBSome

It was previously reported that BBS3 relies on the BBSome for entering cilia in human cells (*Jin et al., 2010*). To investigate whether this applies in *C. reinhardtii*, we examined the BBS1-null *bbs1-1* mutant (*Lechtreck et al., 2009*), in which protein levels of BBS3 and the BBSome subunits BBS4, BBS5, and BBS7 were at wild-type levels (*Figure 2A*; and *Figure 2—figure supplement 1*). These BBSome proteins were not present in cilia of *bbs1-1* cells (*Figure 2B*), and BBS5 was also

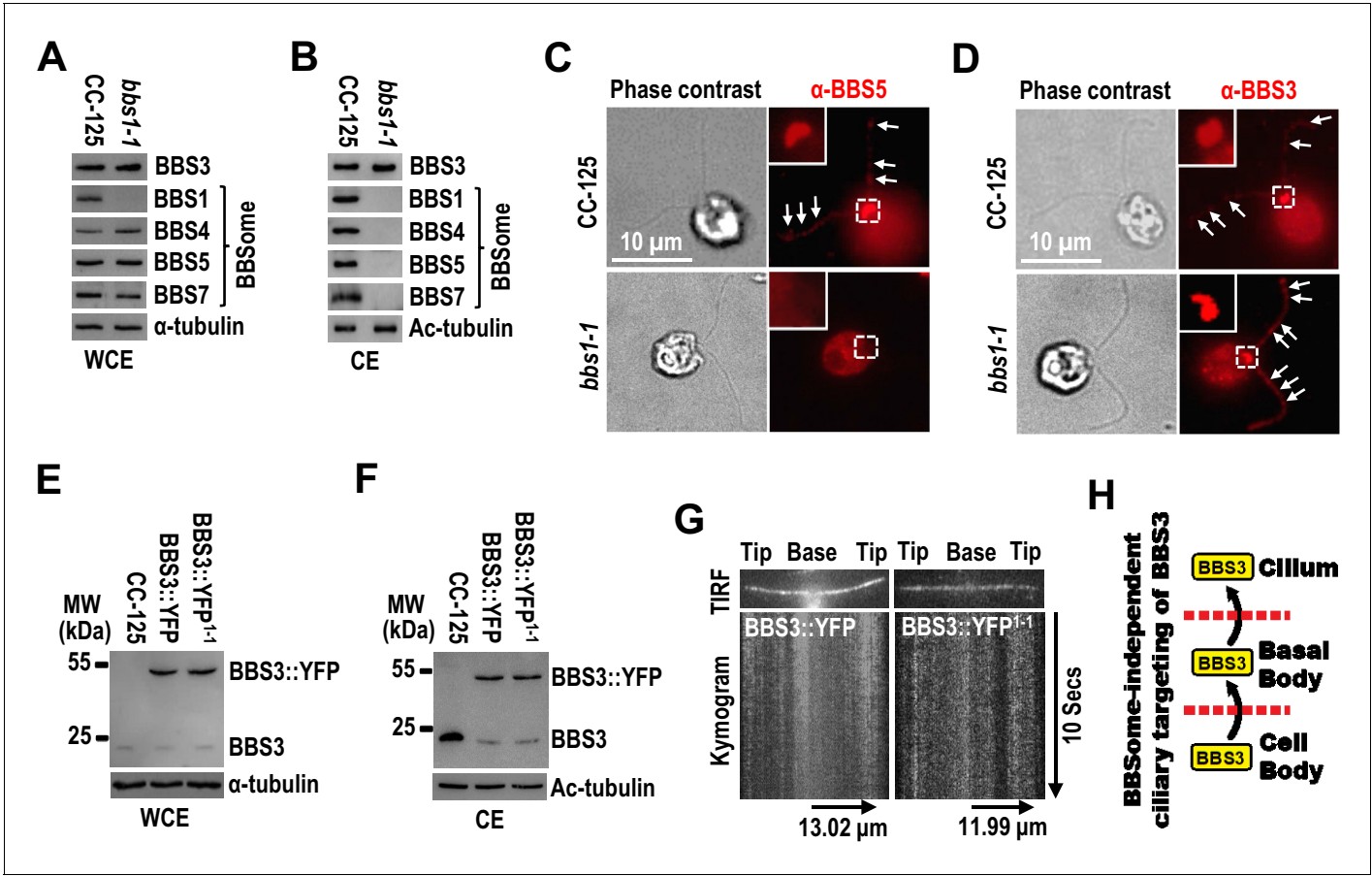

**Figure 2.** BBS3 enters cilia without relying on the BBSome. (A and B) Immunoblots of whole-cell extracts (WCE) (A) and ciliary extracts (CE) (B) of CC-125 and *bbs1-1* cells probed for BBS3 and the BBSome subunits BBS1, BBS4, BBS5, and BBS7. (C and D) CC-125 and *bbs1-1* cells stained with α-BBS5 (red) (C) and α-BBS3 (red) (D). The corresponding phase contrast images are present. Inset shows the basal bodies. White arrows show ciliary staining. Scale bar 10 μm. (E and F) Immunoblots of WCE (E) and CE (F) of CC-125, BBS3::YFP (BBS3::YFP-expressing CC-125 cells), and BBS3::YFP[1-1] (BBS3::YFP-expressing *bbs1-1* cells) cells probed with α-BBS3. (G) Total internal reflection fluorescence images and corresponding kymograms of BBS3::YFP and BBS3::YFP[1-1] cells (*Figure 2—videos 1* and *2*, 15 fps). The time and transport lengths are indicated on the right and on the bottom, respectively. (H) Schematic presentation showing that BBS3 can traffic from the cell body to the basal body for entering cilia in a BBSome-independent manner. For panels A, B, E, and F, α-tubulin and Ac-tubulin were used to adjust the loading of WCE and CE, respectively.

The online version of this article includes the following video and figure supplement(s) for figure 2:

**Figure supplement 1.** Affinity-purified polyclonal antisera against BBS4 and BBS7 can recognize their target proteins specifically.

**Figure 2—video 1.** Total internal reflection fluorescence imaging of BBS3::YFP movement in cilia of CC-125 cells (BBS3::YFP cells).
https://elifesciences.org/articles/59119#fig2video1

**Figure 2—video 2.** Total internal reflection fluorescence imaging of BBS3::YFP movement in cilia of *bbs1-1* cells (BBS3::YFP[1-1] cells).
https://elifesciences.org/articles/59119#fig2video2

absent from the basal bodies in *bbs1-1* cells (*Figure 2C*), indicating that knockout of BBS1 disrupts BBSome assembly and no BBSome is available to be recruited to the basal body for ciliary entry in *bbs1-1* cells. Of note, BBS3 was concentrated at the basal bodies and distributed along the length of cilia in *bbs1-1* cells (*Figure 2D*). Together with the result that BBS3 is present in cilia of *bbs1-1* cells at wild-type level (*Figure 2B*), our data revealed that *Chlamydomonas* BBS3 does not require the BBSome for targeting to the basal body nor for entering cilia. Our previous study revealed that the BBSome loads onto anterograde IFT trains at the basal body and enters cilia via IFT in *C. reinhardtii* (*Xue et al., 2020*). In mammalian cells, BBS3 binds the BBSome through a direct interaction between BBS3 and BBS1 (*Mourão et al., 2014*) and was proposed to enter cilia via IFT through binding the BBSome (*Jin et al., 2010*; *Liew et al., 2014*). Since *Chlamydomonas* BBS3 can enter cilia without relying on the BBSome, we proposed that BBS3 does not enter cilia via IFT in *C. reinhardtii*. Supportive of this notion, when expressed at similar levels in CC-125 and *bbs1-1* cells (resulting strains BBS3::YFP and BBS3::YFP[1-1]) (*Figure 2E*), BBS3::YFP entered cilia of both strains and was maintained in cilia of both strains at similar amounts (*Figure 2F*). As reflected by total internal reflection fluorescence (TIRF) imaging, BBS3::YFP statically distributed along the length of cilia but did not undergo IFT in either strain (*Figure 2G*; and *Figure 2—videos 1* and *2*). Thus, BBS3 can enter cilia independent of the BBSome and does not traffic bidirectionally in cilia via IFT but likely diffuses inside cilia (*Figure 2H*).

## Membrane association is a prerequisite for BBS3 to enter cilia

Other studies have demonstrated that GTP-loaded small GTPase of the Arf family associates with the membrane via its N-terminal amphipathic helix (*Amor et al., 1994*; *Liu et al., 2010*; *Zhang et al., 2011*), and the N-terminal residues 1–15 are essential for BBS3 to associate with the membrane *in vitro* (*Jin et al., 2010*; *Mourão et al., 2014*). When incubated with the synthetic liposomes, bacterially expressed BBS3::YFP associated with liposomes only in the presence of GTPγS, which locks BBS3::YFP in a GTP-bound state (*Figure 3A, B*; *Liew et al., 2014*). In contrast, BBS3△N::YFP, which lacks the N-terminal residues 1–15 of BBS3, did not associate with liposomes even in the presence of GTPγS, revealing that the N-terminal amphipathic helix is required for BBS3 to associate with the membrane, and BBS3 associates with the membrane in a GTP-dependent manner (*Figure 3A, B*). To examine whether the N-terminal residues 1–15 are essential for BBS3 to enter cilia *in vivo*, we expressed BBS3△N::YFP at similar levels in CC-125 and *bbs1-1* cells (resulting strains BBS3△N::YFP and BBS3△N::YFP[1-1]) (*Figure 3C*). In contrast to BBS3::YFP that enters cilia of both CC-125 and *bbs1-1* cells (*Figure 2F, G*), BBS3△N::YFP was absent from cilia of both BBS3△N::YFP and BBS3△N::YFP[1-1] strains (*Figure 3D, E*), indicating that depletion of the N-terminal residues 1–15 prevents BBS3 from entering cilia. Of note, BBS3△N::YFP was concentrated at the basal bodies of both BBS3△N::YFP and BBS3△N::YFP[1-1] strains, revealing that depletion of the N-terminal residues 1–15 retains the ability of BBS3 to traffic from the cell body to the basal body in a BBSome-independent manner (*Figure 3E*). In BBS3△N::YFP[1-1] cells, the BBSome was not assembled due to the lack of BBS1 and thus was not available for targeting to the basal bodies for ciliary entry (*Figure 3D, E*). In contrast, BBS3△N::YFP colocalized with BBS5 at the basal bodies in BBS3△N::YFP cells (*Figure 3E*) and immunoprecipitated the BBSome subunits BBS1 and BBS5 in the cell body extracts of BBS3△N::YFP cells (*Figure 3F*), revealing that BBS3△N::YFP retains the ability to bind and recruit the BBSome to the basal body (*Xue et al., 2020*). Therefore, loss of the N-terminal residues 1–15 of BBS3 does not prevent BBS3 from binding and targeting the BBSome to the basal body, while membrane association is necessary for BBS3 to enter cilia from the basal body in *C. reinhardtii* (*Figure 3G*).

## BBS3 associates with the ciliary membrane in a GTP-dependent manner

BBS3 of several species is unique in that they contain an alanine (73A for *C. reinhardtii*) rather than a glutamine (Q) residue at the position critical for Ras family GTPases to hydrolyze GTP (*Figure 4—figure supplement 1A*). Among them, *Chlamydomonas* BBS3 has been examined to be an active GTPase but with low intrinsic GTPase activity (*Mourão et al., 2014*). Both *in vitro* biochemical and *in vivo* functional experiments demonstrated that BBS3[T31R] is a GDP-locked (inactive) variant (*Figure 4—figure supplement 1A, B*; *Kobayashi et al., 2009*; *Wiens et al., 2010*; *Xue et al., 2020*). A73L mutation disrupted BBS3 GTP hydrolysis, suggesting that BBS3[A73L] is a GTP-locked

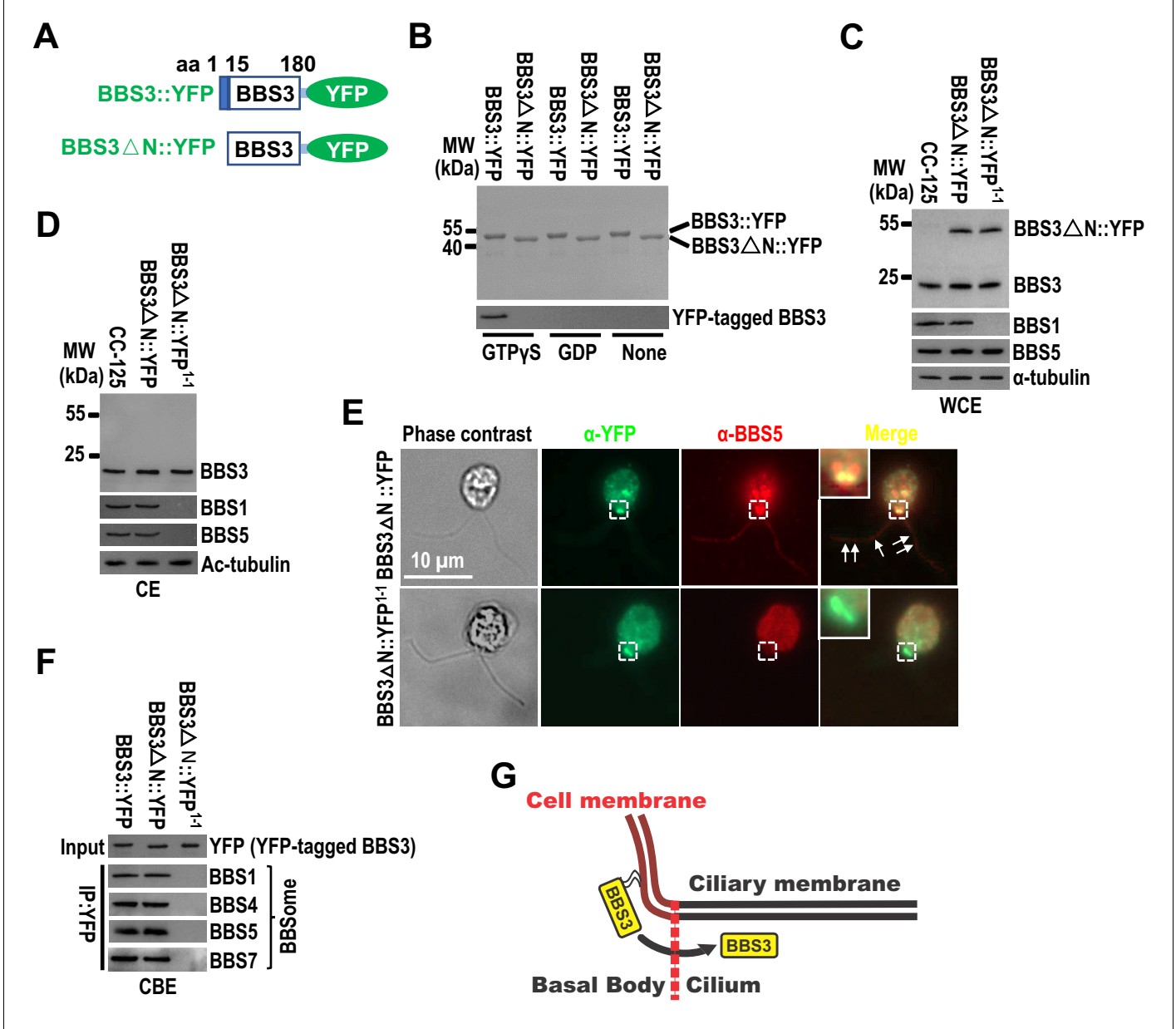

**Figure 3.** Membrane association is a prerequisite for BBS3 to enter cilia. (A) Schematic presentation of BBS3::YFP and BBS3△N::YFP. △N: N-terminal 15 amino acids of BBS3 deleted. (B) SDS-PAGE visualization of bacterially expressed BBS3::YFP and BBS3△N::YFP and immunoblots of liposome-incubated BBS3::YFP and BBS3△N::YFP in the presence of GTPγS, GDP, or neither probed with α-BBS3. (C and D) Immunoblots of whole-cell extracts (WCE) (C) and ciliary extracts (CE) (D) of CC-125, BBS3△N::YFP (BBS3△N::YFP-expressing CC-125 cells), and BBS3△N::YFP[1-1] (BBS3△N::YFP-expressing *bbs1-1* cells) cells probed with α-BBS3, α-BBS1, and α-BBS5. α-tubulin and Ac-tubulin were used to adjust the loading of WCE and CE, respectively. (E) BBS3△N::YFP and BBS3△N::YFP[1-1] cells stained with α-YFP (green) and α-BBS5 (red). The corresponding phase contrast images are present. Inset shows the basal bodies. White arrows show ciliary staining. Scale bar 10 μm. (F) Immunoblots of α-YFP-captured proteins from cell body extracts (CBE) of BBS3::YFP, BBS3△N::YFP, and BBS3△N::YFP[1-1] cells probed with α-BBS1, α-BBS4, α-BBS5, and α-BBS7. Input was quantified with α-YFP by immunoblotting. (G) Schematic presentation of how BBS3 translocates from the basal bodies to cilia on the membrane.

constitutively positive variant (*Figure 4—figure supplement 1A, B*; and *Figure 4—figre supplement 1—source data 1*; *Liew et al., 2014*). Our previous study has revealed that BBS3 in its GTP-bound form (BBS3[A73L]) binds and recruits the BBSome to the basal body for entering cilia (*Xue et al., 2020*). When expressed in CC-125 cells (resulting strain BBS3[A73L]::YFP) (*Figure 4A*), BBS3[A73L]::YFP, like BBS3::YFP, is present in cilia (*Figure 4B, C*) and appears statically distributed

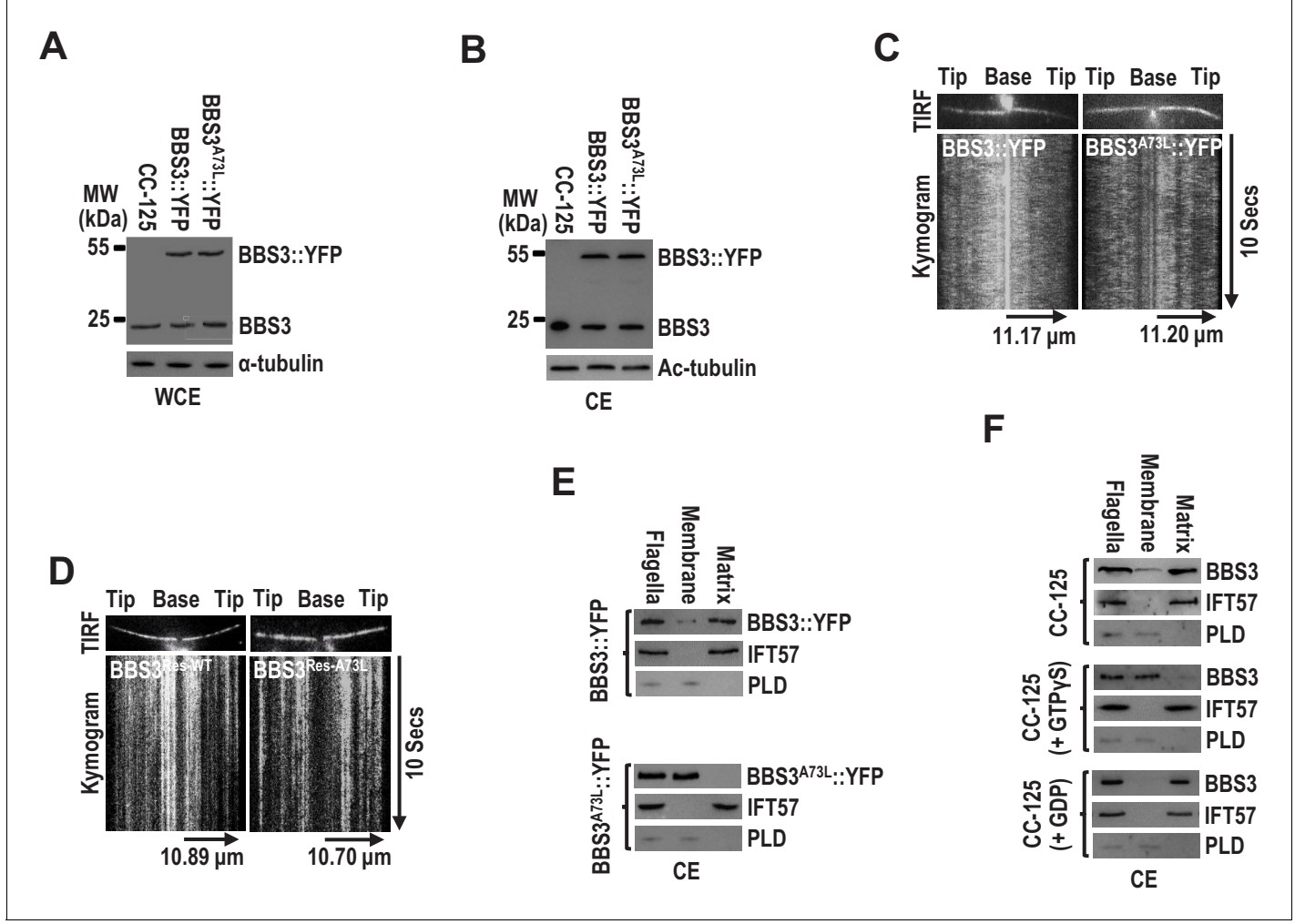

**Figure 4.** BBS3 associates with the ciliary membrane in a GTP-dependent manner. (**A and B**) Immunoblots of whole-cell extracts (WCE) (**A**) and ciliary extracts (CE) (**B**) of CC-125, BBS3::YFP, and BBS3$^{A73L}$::YFP cells probed with α-BBS3. α-tubulin and Ac-tubulin were used to adjust the loading of WCE and CE, respectively. (**C and D**) Total internal reflection fluorescence images and corresponding kymograms of BBS3::YFP and BBS3$^{A73L}$::YFP cells (*Figure 4—videos 1* and *2*, 15 fps) (**C**), and BBS3$^{Res-WT}$ and BBS3$^{Res-A73L}$ cells (*Figure 4—videos 3* and *4*, 15 fps) (**D**). The time and transport lengths are indicated on the right and on the bottom, respectively. (**E**) Immunoblots of flagella and ciliary membrane and matrix fractions of BBS3::YFP and BBS3$^{A73L}$::YFP cells probed with α-BBS3, α-IFT57, and α-PLD. (**F**) Immunoblots of flagella and ciliary membrane and matrix fractions of CC-125 cells, in the presence of GTPγS, GDP, or neither, probed for BBS3, IFT57, and PLD. For panels E and F, IFT57 is solely present in the ciliary matrix fraction, while PLD is only detected in the ciliary membrane fraction.

The online version of this article includes the following video, source data, and figure supplement(s) for figure 4:

**Figure supplement 1.** BBS3 is a highly conserved protein across ciliated species.

**Figure supplement 1—source data 1.** Source data for the GTP hydrolysis experiment shown in *Figure 4—figure supplement 1B*.

**Figure supplement 2.** Affinity-purified polyclonal antisera against PLD can recognize its target protein specifically.

**Figure 4—video 1.** Total internal reflection fluorescence imaging of BBS3::YFP movement in cilia of CC-125 cells (BBS3::YFP cells).
https://elifesciences.org/articles/59119#fig4video1

**Figure 4—video 2.** Total internal reflection fluorescence imaging of BBS3$^{A73L}$::YFP movement in cilia of CC-125 cells (BBS3$^{A73L}$::YFP cells).
https://elifesciences.org/articles/59119#fig4video2

**Figure 4—video 3.** Total internal reflection fluorescence imaging of BBS3::GFP movement in cilia of BBS3$^{miRNA}$ cells (BBS3$^{Res-WT}$ cells).
https://elifesciences.org/articles/59119#fig4video3

**Figure 4—video 4.** Total internal reflection fluorescence imaging of BBS3$^{A73L}$::GFP movement in cilia of BBS3$^{miRNA}$ cells (BBS3$^{Res-A73L}$ cells).
https://elifesciences.org/articles/59119#fig4video4

along the length of cilia as reflected by TIRF imaging (*Figure 4C*; and *Figure 4—videos 1* and *2*). Similar stationary pattern was also recorded for the C-terminal green fluorescent protein (GFP)-tagged BBS3 and the A73L mutant in rescuing strains BBS3[Res-WT] and BBS3[Res-A73L], in which BBS3::GFP and BBS3[A73L]::GFP are expressed in similar amounts in the BBS3-knockdown strain BBS3[miRNA] (*Figure 4D*; and *Figure 4—videos 3* and *4*; *Xue et al., 2020*). Upon entering cilia, BBS3[A73L]::YFP was detected in the membrane fraction but not in the matrix fraction isolated from cilia of the BBS3[A73L]::YFP strain, while the majority of BBS3::YFP was present in the matrix faction of cilia of the BBS3::YFP strain (*Figure 4E*; and *Figure 4—figure supplement 2*). To confirm this observation, we performed ciliary fraction analysis on CC-125 cells in the presence of GTPγS, GDP, or neither, which are proposed to lock BBS3 at a GTP-bound, GDP-bound, or wild-type state, respectively (*Liew et al., 2014*). Our data showed that BBS3 is mainly present in the matrix fraction in the absence of nucleotide or completely present in the matrix fraction when GDP was added, while the majority of BBS3 was detected in the membrane faction in the presence of GTPγS (*Figure 4F*). Therefore, GTP binding allows BBS3 to attach to the ciliary membrane, and GDP binding detaches BBS3 from the ciliary membrane in cilia, consistent with the observation that GTP- but not GDP-bound BBS3 associates with the membrane of liposomes *in vitro* (*Figure 3B*; *Mourão et al., 2014*).

## BBS3 interacts with the BBSome in cilia in a GTP-dependent manner

Rodent BBS3, in its GTP-bound state, binds and recruits the BBSome to the membrane of liposomes *in vitro*, implying that, in cilia, GTP loading confers BBS3 to bring the BBSome to the ciliary membrane (*Liew et al., 2014*). To investigate whether and how BBS3 interacts with the BBSome in cilia of *C. reinhardtii* cells, we performed sucrose gradient density centrifugation on the ciliary extracts of BBS3::YFP and BBS3[A73L]::YFP transgenic strains. Of note, a minority of BBS3::YFP co-sedimented with the BBSome subunits BBS1 and BBS5 (*Figure 5A*). In contrast, the majority of BBS3[A73L]::YFP co-sedimented with the two BBSome proteins although partial BBS3[A73L]::YFP remained to be existing as a free form independent of the two BBSome proteins in cilia (*Figure 5A*). These results showed that GTP-loaded BBS3 is able to bind the BBSome in cilia (*Figure 5A*). This notion was confirmed by observing that the BBS3[A73L] variant but not BBS3::YFP recovered BBS1 and BBS5 in the ciliary extracts (*Figure 5B*). To further verify this result, we added GTPγS or GDP, which is proposed to lock GTPases in a GTP- or GDP-bound state, respectively, to the ciliary extracts of BBS3::YFP cells (*Liew et al., 2014*). Sucrose gradient density centrifugation assay identified BBS3::YFP sediments separately from BBS1 and BBS5 in the presence of GDP but became partially co-sedimented with the two BBSome proteins when GTPγS was present (*Figure 5C*). Together with the observation that BBS3::YFP immunoprecipitated BBS1 and BBS5 in the ciliary extracts of the BBS3::YFP transgenic strain only when GTPγS was present (*Figure 5D*), we concluded that BBS3 in a GTP- but not GDP-bound configuration binds the BBSome in cilia. Of note, partial BBS3, even when locked in a GTP-bound configuration, remained not to interact with the BBSome, revealing that GTP loading cannot confer all BBS3 molecules to interact with the BBSome in cilia (*Figure 5A, C*).

## The BBSome cycles through cilia normally in the absence of BBS3 in cilia

Once at the ciliary tip, the BBSome/IFT train is proposed to remodel before undergoing turnaround to exit cilia (*Wei et al., 2012*). During this process, rodent BBS3 was reported to mediate BBSome exit out of cilia by promoting its loading onto retrograde IFT trains at the ciliary tip (*Liew et al., 2014*). To investigate whether this applies in *C. reinhardtii*, we examined *clip1* cells and found that loss of BBS3 in cilia did not cause ciliary hyperaccumulation of the BBSome subunits BBS1, BBS4, BBS5, and BBS7, excluding BBS3 from being required for the BBSome to exit cilia in *C. reinhardtii* (*Figure 6A* and *Figure 1B*; *Liew et al., 2014*). Our previous study has shown that loss of IFT25 blocks loading of the BBSome onto retrograde IFT trains (*Dong et al., 2017b*). Reflecting this observation, BBS1 and BBS5 hyperaccumulated at the ciliary tip in the IFT25-knockdown cells (*Figure 6B*). BBS1 and BBS5 co-sedimented with each other and peaked at the same fraction as IFT-B1 (checked with IFT46 and IFT70) in the ciliary extracts of wild-type CC-5325 cells (*Figure 6C*; *Xue et al., 2020*). In contrast, these two BBSome proteins rather than IFT-B1 were found to shift to lower fractions in the ciliary extracts of IFT25[miRNA] cells, suggesting that they are not able to assemble to form intact BBSomes after remodeling at the ciliary tip (*Figure 6C*). As compared to the IFT25-knockdown cells, *clip1* cells did not accumulate BBS1 and BBS5 at the ciliary tip and the two BBSome subunit proteins

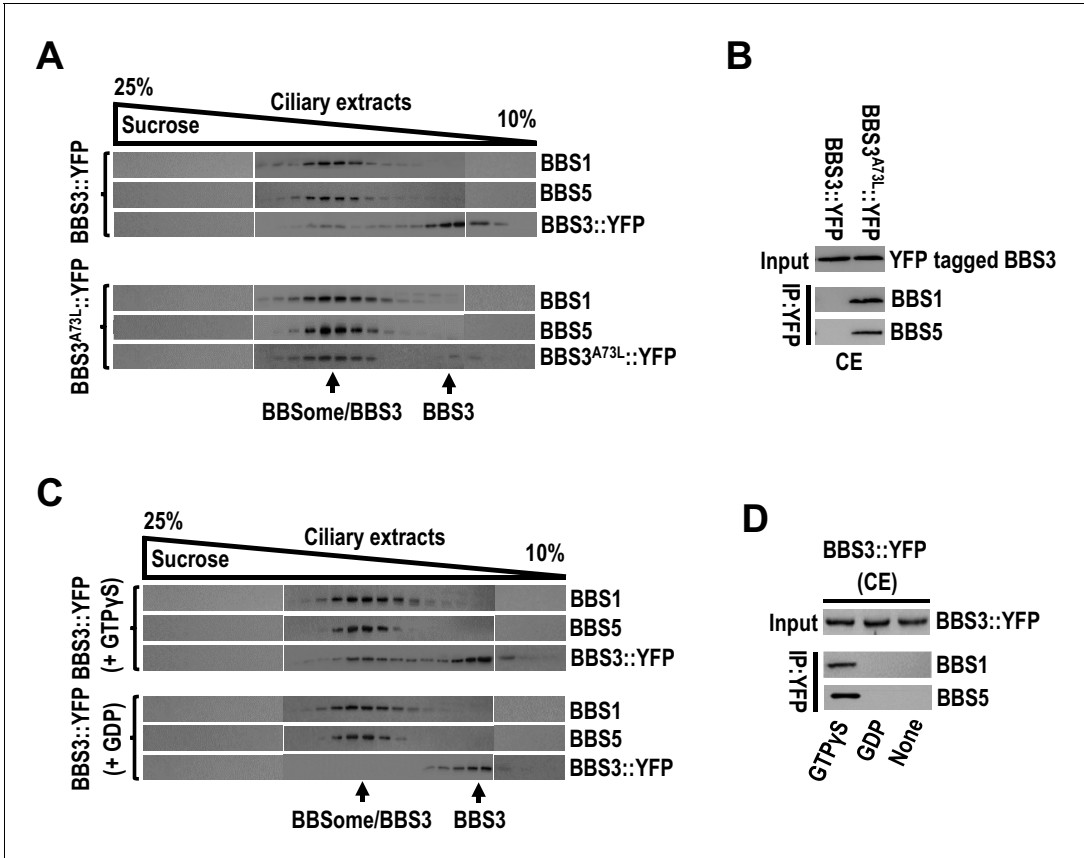

**Figure 5.** BBS3 interacts with the BBSome in cilia in a GTP-dependent manner. (**A**) Immunoblots of sucrose density gradient of ciliary extracts (CE) of BBS3::YFP and BBS3^{A73L}::YFP cells probed with α-BBS1, α-BBS5, and α-YFP. (**B**) Immunoblots of α-YFP-captured proteins from CE of BBS3^{A73L}::YFP and BBS3::YFP cells probed with α-BBS1 and α-BBS5. (**C**) Immunoblots of sucrose density gradient of CE of BBS3::YFP cells, in the presence of GTPγS or GDP, probed with α-BBS1, α-BBS5, and α-GFP. (**D**) Immunoblots of α-YFP-captured proteins from CE of BBS3::YFP cells, in the presence of GTPγS, GDP, or neither, probed with α-BBS1 and α-BBS5. For panels B and D, input was quantified with α-YFP by immunoblotting.

co-sedimented in the same fractions as IFT-B1 in the ciliary extracts of *clip1* cells, demonstrating that ciliary dynamics of the BBSome is normally maintained when BBS3 is absent from cilia (*Figure 6B, C*). These results further argued against BBS3 being required for the BBSome to load onto retrograde IFT trains for ciliary exit (*Liew et al., 2014*). Next, we expressed BBS5::YFP in similar amounts in CC-5325 and *clip1* cells (resulting strains BBS5::YFP^{5325} and BBS5::YFP^{clip1}) (*Figure 6D*). BBS5::YFP entered cilia of both strains and was maintained at similar amounts (*Figure 6E*). As determined by TIRF imaging, BBS5::YFP underwent IFT with normal speeds and velocities in cilia of *clip1* cells, suggesting that IFT of the BBSome is not impaired when BBS3 is absent from cilia (*Figure 6F, G*; *Figure 6—source data 1*; and *Figure 6—videos 1* and *2*). Therefore, we concluded that BBS3 is not required for maintaining BBSome dynamics in cilia, nor for promoting BBSome coupling/uncoupling with IFT trains at the ciliary tip in *C. reinhardtii* (*Figure 6H*).

## BBS3 is essential for PLD to associate with the BBSome for ciliary exit

In *C. reinhardtii*, the membrane-associated signaling protein PLD enters cilia mostly by diffusion but is removed from cilia mainly via IFT by associating with the BBSome. Therefore, the *bbs* mutants accumulates PLD in cilia as the BBSome is not available for entering cilia to bridge PLD to retrograde IFT trains for ciliary exit in those mutants (*Lechtreck et al., 2013*; *Liu and Lechtreck, 2018*). Of note, the *clip1* strain contained PLD at wild-type level in whole-cell sample (*Figure 7A*) and accumulated PLD in cilia, mostly at the ciliary tip (*Figure 7B, C*), revealing that BBS3 is essential for promoting PLD to exit cilia. In *clip1* cells, BBS3 was maintained at wild-type level in whole-cell sample (*Figure 7A*) but was absent from cilia (*Figure 7B*), and its ciliary absence did not affect BBSome

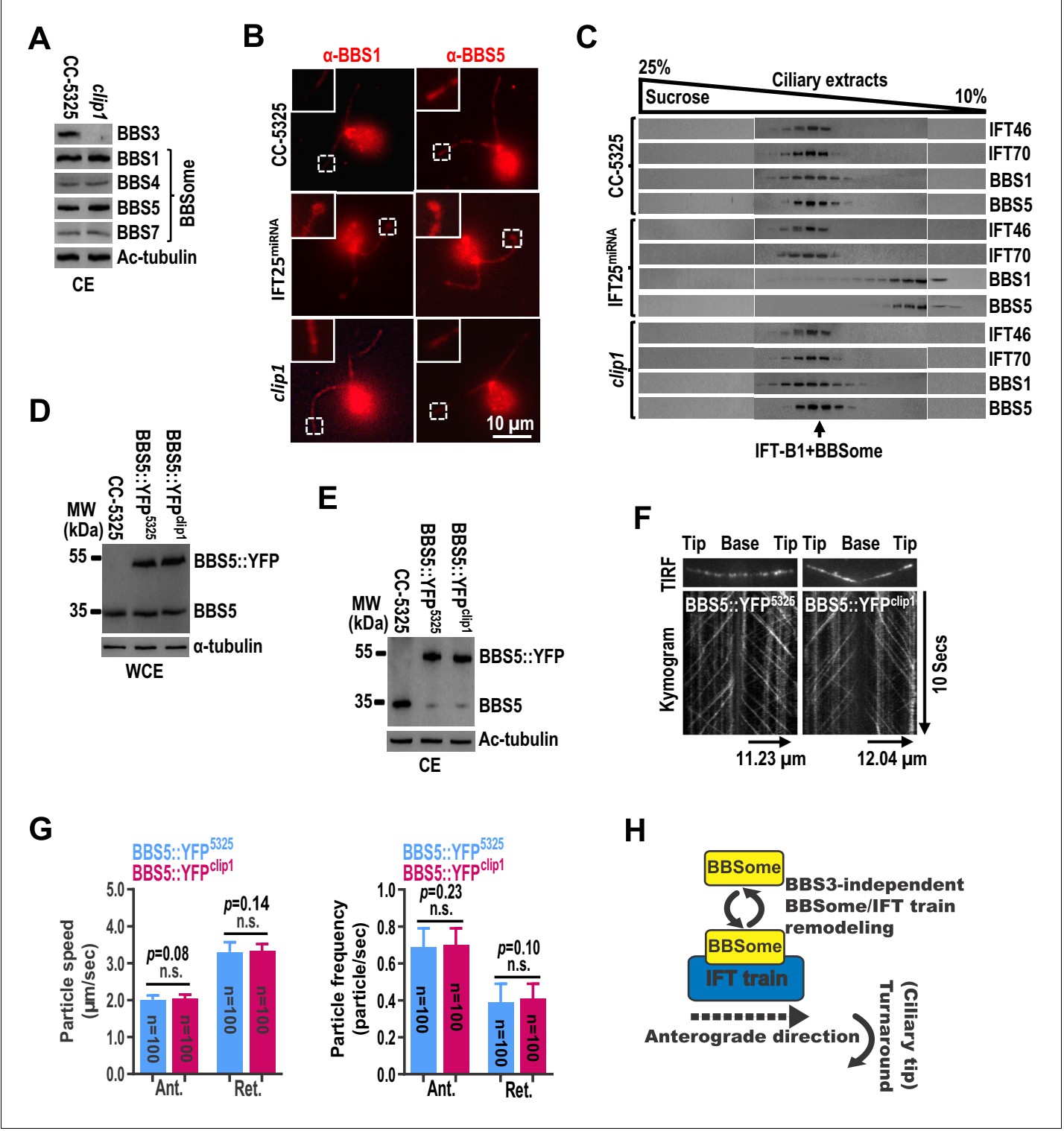

**Figure 6.** The BBSome cycles through cilia normally in the absence of BBS3 in cilia. (**A**) Immunoblots of ciliary extracts (CE) of CC-5325 and *clip1* cells probed for BBS3 and the BBSome subunits BBS1, BBS4, BBS5, and BBS7. Ac-tubulin was used to adjust the loading. (**B**) CC-5325, IFT25^miRNA, and *clip1* cells stained with α-BBS1 (red) and α-BBS5 (red). Inset shows ciliary tips. Scale bar 10 µm. (**C**) Immunoblots of sucrose density gradient of CE of CC-5325, IFT25^miRNA, and *clip1* probed with α-IFT46, α-IFT70, α-BBS1, and α-BBS5. (**D and E**) Immunoblots of whole-cell extracts (WCE) (**D**) and CE (**E**) of CC-5325, BBS5::YFP^5325, and BBS5::YFP^clip1 cells probed with α-BBS5. α-tubulin and Ac-tubulin were used to adjust the loading of WCE and CE, respectively. (**F**) Total internal reflection fluorescence images and corresponding kymograms of BBS5::YFP^5325 and BBS5::YFP^clip1 cells (*Figure 6— videos 1* and *2*, 15 fps). The time and transport lengths are indicated on the right and on the bottom, respectively. (**G**) Speeds and frequencies of

*Figure 6 continued on next page*

*Figure 6 continued*

BBS5::YFP inside cilia of BBS5::YFP$^{5325}$ and BBS5::YFP$^{clip1}$ cells are similar. The speeds of BBS5::YFP were 2.01 ± 0.12 (anterograde) and 3.31 ± 0.27 μm/sec (retrograde), and the frequencies were 0.69 ± 0.10 (anterograde) and 0.39 ± 0.10 particles/sec (retrograde) for BBS5::YFP$^{5325}$ cells. In cilia of BBS5::YFP$^{clip1}$ cells, the speeds of BBS5::YFP were 2.04 ± 0.11 (anterograde) and 3.34 ± 0.19 μm/sec (retrograde), and the frequencies were 0.70 ± 0.09 (anterograde) and 0.41 ± 0.08 particles/sec (retrograde). Error bar indicates SD. n: number of cilia analyzed; n.s.: non-significance. (**H**) Schematic presentation showing that the BBSome and IFT trains remodel at the ciliary tip in a BBS3-independent manner.

The online version of this article includes the following video and source data for figure 6:

**Source data 1.** Source data for the speed and frequency experiment shown in *Figure 6G*.

**Figure 6—video 1.** Total internal reflection fluorescence imaging of BBS5::YFP movement in cilia of CC-5325 cells (BBS5::YFP$^{5325}$ cells).
https://elifesciences.org/articles/59119#fig6video1

**Figure 6—video 2.** Total internal reflection fluorescence imaging of BBS5::YFP movement in cilia of *clip1* cells (BBS5::YFP$^{clip1}$ cells).
https://elifesciences.org/articles/59119#fig6video2

abundance in cilia and uncoupling/recoupling of the BBSome with IFT at the ciliary tip (*Figure 6A–F*); we thus concluded that *Chlamydomonas* BBS3 is essential for PLD to associate with the BBSome at the ciliary tip for ciliary exit via IFT. To further verify this notion, we examined the BBS3-knockdown strain, BBS3$^{miRNA}$, in which the BBSome (checked with BBS1 and BBS5) was maintained at wild-type level in whole-cell sample but was strongly reduced in cilia (*Figure 7D, E*; *Xue et al., 2020*). BBS3 knockdown did not alter cellular PLD level but accumulated PLD in cilia, mostly at the ciliary tip (*Figure 7D–F*). When BBS3::GFP was expressed in BBS3$^{miRNA}$ cells (resulting strain BBS3$^{Res-WT}$), it entered cilia and restored BBS1, BBS5, and PLD to wild-type levels (*Figure 7D–F*; *Xue et al., 2020*). When BBS3△N::YFP was expressed in BBS3$^{miRNA}$ cells (resulting strain BBS3$^{Res-△N}$), BBS1 and BBS5 were rescued to wild-type levels in cilia, while PLD remained to be accumulated mostly at the ciliary tip (*Figure 7D–F*). Since BBS3 was absent from cilia of both BBS3$^{miRNA}$ and BBS3$^{Res-△N}$ cells and BBS1 and BBS5 were restored to wild-type levels in cilia of BBS3$^{Res-△N}$ cells, our data thus showed that ciliary absence of BBS3 causes PLD to accumulate mostly at the ciliary tip even when intact BBSomes are present in cilia (*Figure 7E–G*), confirming that BBS3 is essential for PLD to associate with the BBSome at the ciliary tip for ciliary exit. Remarkably, the BBS3$^{A73L}$::GFP-expressing BBS3$^{miRNA}$ strain BBS3$^{Res-A73L}$ also restored the ciliary PLD content to wild-type level, excluding GTPase cycling of BBS3 in cilia from mediating PLD association with the BBSome for ciliary exit (*Figure 7D–F*). Together with the observation that PLD rarely co-sedimented with the BBSome (checked with BBS1 and BBS5) in the ciliary extracts of *clip1* cells; a minority of PLD co-sedimented with the BBSome in the ciliary extracts of BBS3$^{Res-△N}$ cells; and the majority of PLD became co-sedimented with the BBSome in the ciliary extracts of BBS3$^{Res-A73L}$ cells (*Figure 7H*), our data suggest that GTP-bound BBS3 efficiently enables PLD to associate with the BBSome in cilia at the ciliary tip to undergo retrograde IFT (*Figure 7I*). In addition, BBS3△N::YFP did not enter cilia itself but restored BBS1 and BBS5 to wild-type levels in cilia of the BBS3-knockdown BBS3$^{miRNA}$ cells (*Figure 7D–F*), providing compelling evidence to confirm that ciliary entry of the BBSome from the ciliary base does not depend on BBS3 (*Figure 1*).

## Discussion

Using *C. reinhardtii* as a model organism, we elucidated the interplay between BBS3 and the BBSome for ciliary entry and investigated their dynamics in cilia and the role of BBS3 in promoting ciliary exit of the signaling protein PLD via the BBSome. Our data show that BBS3 plays a crucial role in mediating ciliary exit of PLD through promoting its association with the BBSome at the ciliary tip, thus closing a gap in our understanding of the role of BBS3 in regulating ciliary exit of signaling protein cargoes.

### BBS3 and the BBSome interplay non-reciprocally for targeting to the basal body

The BBSome is the major effector of BBS3 in human cells (*Jin et al., 2010*). BBS3 binds the BBSome through a direct interaction with the BBSome subunit BBS1 in human, mouse, and *Chlamydomonas* cells (*Mourão et al., 2014*; *Zhang et al., 2012*). In the cell body of *C. reinhardtii*, this binding does not rely on the BBS3 nucleotide state; however, only GTP-bound BBS3 recruits the BBSome to the

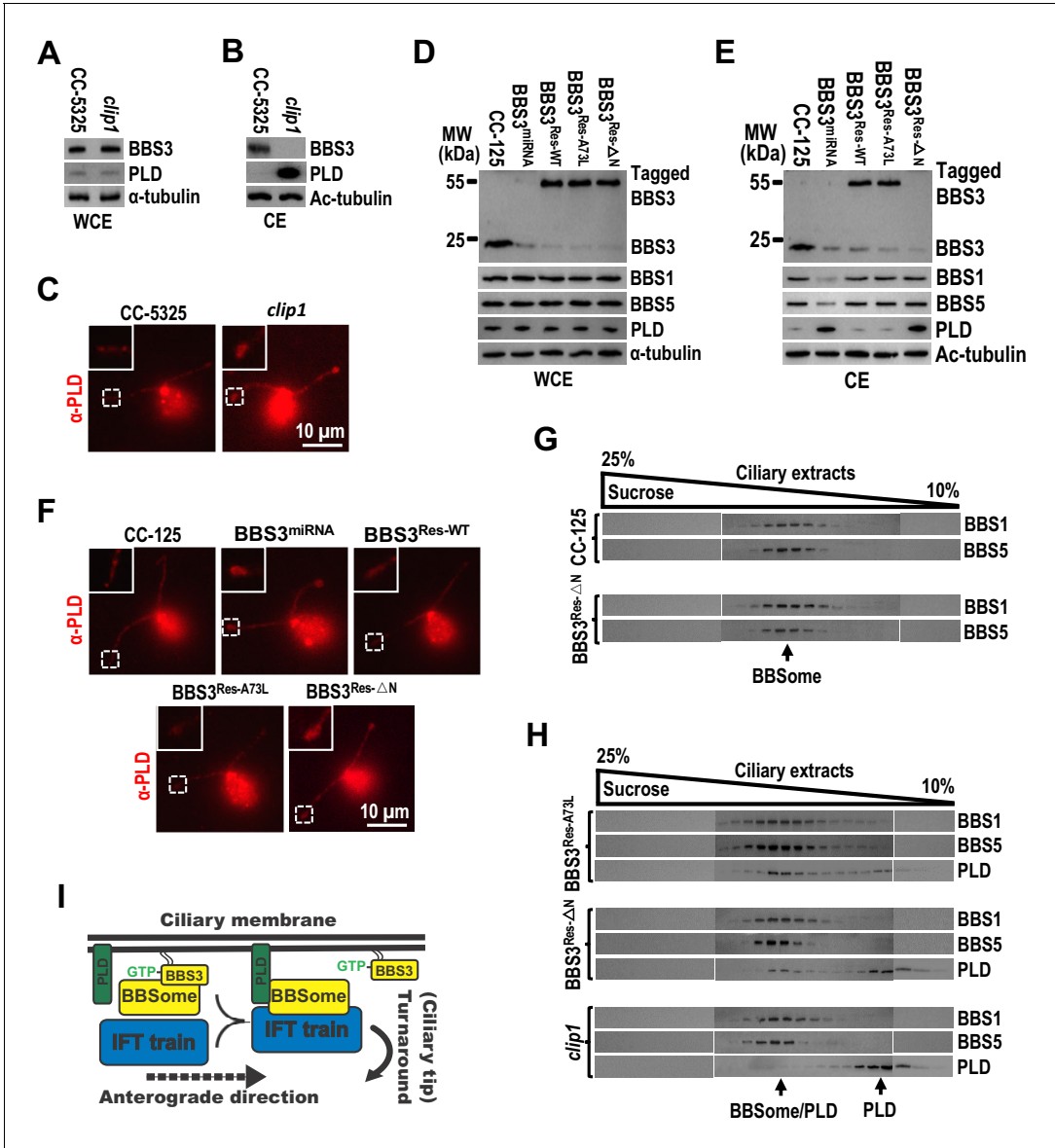

**Figure 7.** BBS3 is essential for PLD to associate with the BBSome for ciliary exit. (**A and B**) Immunoblots of whole-cell extracts (WCE) (**A**) and ciliary extracts (CE) (**B**) of CC-5325 and *clip1* cells probed with α-BBS3 and α-PLD. (**C**) CC-5325 and *clip1* cells stained with α-PLD (red). (**D and E**) Immunoblots of WCE (**D**) and CE (**E**) of CC-125, BBS3^miRNA, BBS3^Res-WT, BBS3^Res-A73L, and BBS3^Res-△N cells probed with α-BBS3, α-BBS1, α-BBS5, and α-PLD. (**F**) CC-125, BBS3^miRNA, BBS3^Res-WT, BBS3^Res-A73L, and BBS3^Res-△N cells stained with α-PLD (red). (**G**) Immunoblots of sucrose density gradient of CE of CC-125 and BBS3^Res-△N cells probed with α-BBS1 and α-BBS5. (**H**) Immunoblots of sucrose density gradient of CE of BBS3^Res-A73L, BBS3^Res-△N, and *clip1* cells probed with α-BBS1, α-BBS5, and α-PLD. (**I**) Schematic presentation of how BBS3 binds and recruits the BBSome to the ciliary membrane for interacting with PLD and how PLD-associated BBSome loads onto IFT trains for ciliary exit. For panels A, B, D, and E, α-tubulin and Ac-tubulin were used to adjust the loading for WCE and CE, respectively. For panels C and F, the inset shows ciliary tips. Scale bar 10 μm.

basal body, suggesting that basal body recruitment of the BBSome is regulated by the nucleotide state of BBS3 (*Xue et al., 2020*). Such a scenario would allow cells to control the amount of BBSomes available for pick-up by anterograde IFT trains at the basal body and in turn could regulate the presence and amount of signaling proteins in cilia, which rely on the BBSome for ciliary export and import via IFT (*Wingfield et al., 2018*; *Xue et al., 2020*). Indeed, basal body-associated pools have been also described for other IFT cargoes (*Dai et al., 2018*). Of note, the BBSome relies on GTP-bound BBS3 to target to the basal body, while GTP-bound BBS3 concentrates at the basal body even in the absence of the BBSome, suggesting that BBS3 does not rely on the BBSome for

trafficking from the cell body to the basal body in *C. reinhardtii*. Thus, unlike the observation that GTP-bound BBS3 and the BBSome target to the ciliary base of human cells in an interdependent manner (*Jin et al., 2010*), the BBSome relies on GTP-bound BBS3 to target to the basal body but not *vice versa* in *C. reinhardtii*.

## BBS3 and the BBSome translocate from the basal body to cilia independently

Upon targeting to the basal body, the BBSome loads onto anterograde IFT trains through associating with IFT-B1 to enter cilia (*Xue et al., 2020*). An analysis of both the *clip1* mutant and the BBS3-knockdown strain BBS3[miRNA] shows that this happens even when BBS3 is not able to enter cilia. Furthermore, BBS3 translocates from the basal body to cilia even in the absence of the BBSome. These results reveal that BBS3 and the BBSome do not rely on each other for translocating from the basal body to cilia in *C. reinhardtii*. Loss of the N-terminal amphipathic helix essential for BBS3 to associate with the membrane prevents BBS3 from translocating from the basal body to cilia but has no effect on the ability of BBS3 to traffic from the cell body to the basal body. These results reveal that membrane association is required for BBS3 to translocate from the basal body to cilia but not for BBS3 to traffic from the cell body to the basal body. Thus, our data provide *in vivo* evidence, for the first time, to show that BBS3 enters cilia from the basal body potentially by lateral transport on the membrane, consistent with the notion previously proposed for rodent cells (*Jin et al., 2010*; *Liew et al., 2014*; *Mourão et al., 2014*; *Zhang et al., 2011*). Our previous study has shown that RABL5/IFT22 binds and stabilizes BBS3 independent of their nucleotide states in the cell body (*Xue et al., 2020*). Since the GTP-bound configuration is related to BBS3's ability to associate with the membrane (*Jin et al., 2010*; *Klink et al., 2020*; *Liew et al., 2014*; *Mourão et al., 2014*; *Singh et al., 2020*; *Zhang et al., 2011*), IFT22 binding to BBS3 in the cell body could prevent BBS3 from associating with the cell membrane. Before ciliary entry, IFT22 is released from BBS3 at the basal body, which will enable the GTP-bound BBS3 to be in a state to attach to the membrane for lateral transport into cilia (*Figure 8*). Therefore, the IFT22-dependent recruitment of BBS3 to the basal body and then separation from BBS3 could also function to regulate BBS3 activity at the basal body, ensuring a spatial restriction of its association with the membrane to the basal body and ciliary compartment (*Xue et al., 2020*).

## Where and how does BBS3 bind the BBSome in cilia?

Unlike its counterparts in *Caenorhabditis elegans* and mammalian olfactory neurons that undergo IFT in cilia (*Fan et al., 2004*; *Williams et al., 2014*), *Chlamydomonas* BBS3, upon entering cilia, diffuses between the ciliary base and tip (*Figure 4C, D*). This excludes BBS3 from coupling with BBSomes during its transportation between the ciliary base and tip, a notion also supported by the observation that partial BBS3 molecules remain to be separated from the BBSome in cilia even in its GTP-locked configuration (BBS3[A73L]) or in the presence of GTPγS (*Figure 5A, C*). However, GTP-bound BBS3 recovers BBSomes in cilia by immunoprecipitation, and a majority of GTP-locked BBS3 co-sediments with BBSomes in cilia (*Figure 5A–D*), revealing that GTP-bound BBS3 binds BBSomes most likely at the ciliary tip. These observations demonstrate that GTP loading does not confer BBS3 to bind BBSomes during its transportation between the ciliary base and tip. While at the ciliary tip, BBS3, however, gains the ability to bind BBSomes in a GTP-dependent manner, in agreement with the report that GTP loading confers BBS3 to bind BBSomes in cilia of human cells (*Jin et al., 2010*). Why GTP-locked BBS3 selects to bind or not to bind the BBSome in different ciliary compartments remains unknown at present, and the underlying molecular mechanism deserves to be further investigated (*Figure 8*).

## BBS3 likely promotes the association of PLD with the BBSome at the ciliary tip

BBS3 is assumed to convert to and exist as a GDP-bound configuration upon entering cilia in rodent cells (*Liew et al., 2014*). It was also proposed that BBS3 undergoes a GTPase cycle at the ciliary tip, and RABL4/IFT27 activates GDP-bound BBS3 as a BBS3-specific GEF during this process (*Liew et al., 2014*). In rodent cells, GTP-bound BBS3 promotes the cargo-laden BBSome to reload onto retrograde IFT trains at the ciliary tip for ciliary exit via IFT (*Liew et al., 2014*). In *C. reinhardtii*,

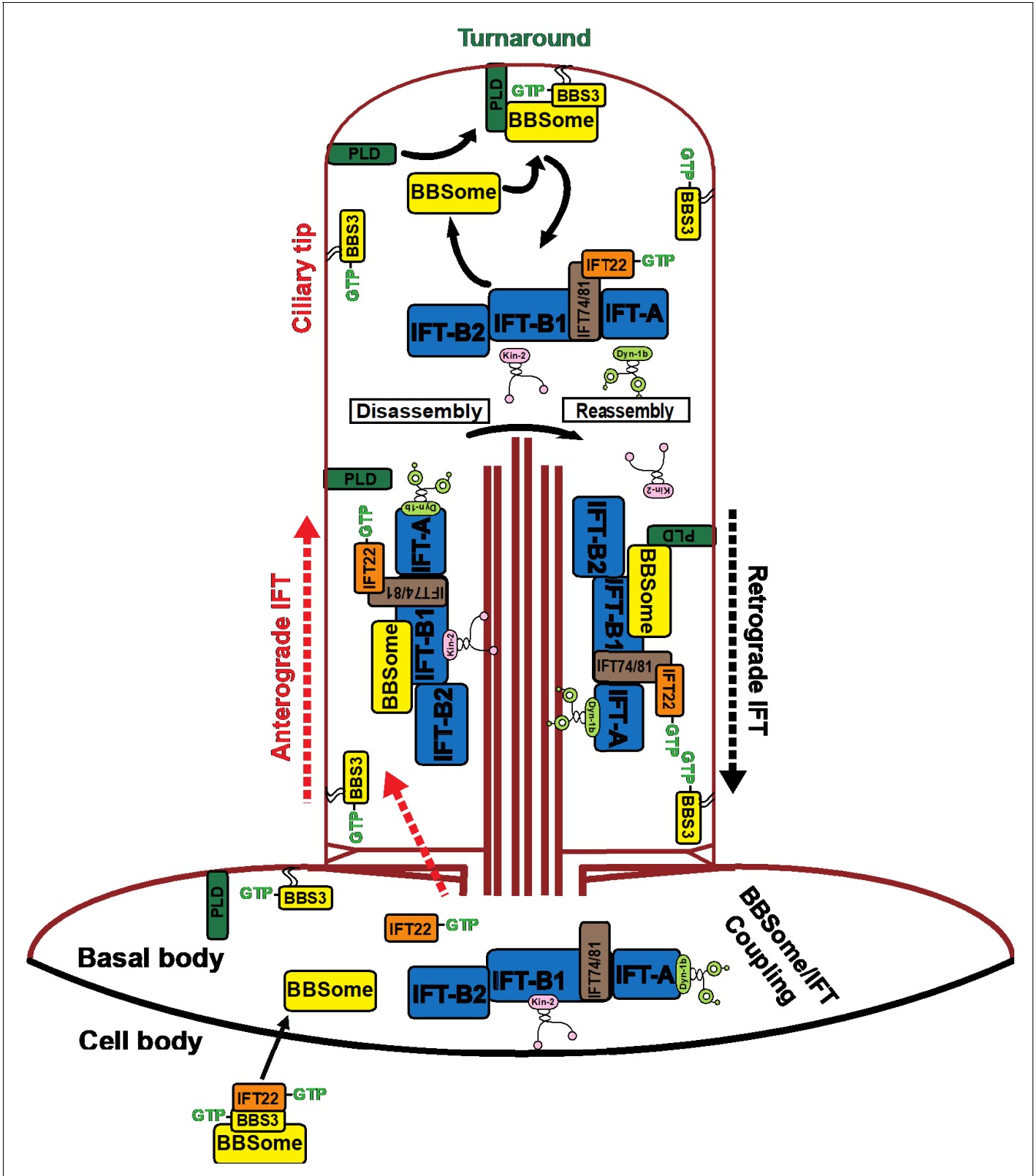

**Figure 8.** Hypothetical model of how BBS3 enters cilia to promote ciliary exit of PLD via the BBSome in *C. reinhardtii*. IFT22 and BBS3 both in a GTP-bound configuration recruit the BBSome to the basal body, where the BBSome separates from BBS3 and loads onto the anterograde IFT trains by coupling with IFT-B1 and enters cilia via IFT (**Xue et al., 2020**). GTP-bound IFT22 is released from BBS3 and binds IFT-B1 through a direct interaction between IFT22 and the IFT74/81 sub-complex for ciliary entry (**Bhogaraju et al., 2013**; **Lucker et al., 2010**; **Taschner et al., 2014**). GTP-bound BBS3

*Figure 8 continued on next page*

*Figure 8 continued*

then associates with the cell membrane to diffuse into cilia. PLD enters cilia by diffusion on membrane (*Liu and Lechtreck, 2018*). At the ciliary tip, the BBSome is released from anterograde IFT trains through BBSome/IFT train remodeling (*Xue et al., 2020*). GTP-bound BBS3 binds and recruits the BBSome to the ciliary membrane for interacting with PLD (*Liu and Lechtreck, 2018*). PLD-laden BBSomes then load onto retrograde IFT trains by coupling with IFT-B1 for ciliary exit (*Xue et al., 2020*). The kinesin-II anterograde motor and the cytoplasmic dynein-1b retrograde motor are also shown (*Cole et al., 1998*; *Pazour et al., 1998*). After turnaround at the ciliary tip, kinesin-II diffuses back to the ciliary base (*Chien et al., 2017*; *Hendel et al., 2018*).

an analysis of the mutant strains indicates that the BBSome migrates from the basal body to cilia and cycles through cilia normally in the absence of BBS3, excluding BBS3 from mediating loading of the BBSome onto retrograde IFT trains for ciliary exit at the ciliary tip. Instead, GTP-bound BBS3 likely couples with the ciliary membrane and binds BBSomes simultaneously at the ciliary tip. This could recruit BBSomes to the ciliary membrane for interacting with PLD (*Figure 8*). Therefore, BBS3 is proposed to participate in mediating the association of ciliary signaling cargoes, for example, PLD, with BBSomes at the ciliary tip (*Jin et al., 2010*; *Klink et al., 2020*; *Klink et al., 2017*; *Liew et al., 2014*; *Mourão et al., 2014*). Interestingly, a BBS3$^{A73L}$ mutant can rescue the accumulated PLD to wild-type level in cilia, suggesting that, at least in cilia of *Chlamydomonas* cells, BBS3 that is locked in a GTP-bound configuration is able to promote ciliary signaling molecules to associate with BBSomes for ciliary exit via IFT. This eventually excludes BBS3 GTPase cycling at the ciliary tip from being involved in promoting PLD to exit cilia in *C. reinhardtii* (*Liew et al., 2014*).

# Materials and methods

## Key resources table

| Reagent type (species) or resource | Designation | Source or reference | Identifiers | Additional information |
|---|---|---|---|---|
| Genetic reagent (*C. reinhardtii*) | CC-125 | *Chlamydomonas* Genetic Center (CGC) | CC-125 | Wild-type strain |
| Genetic reagent (*C. reinhardtii*) | CC-5325 | CGC | CC-5325 | Wild-type strain |
| Genetic reagent (*C. reinhardtii*) | *clip1* | CGC | LMJ.RY0402.149010 | CLiP mutant/CC-5323 |
| Genetic reagent (*C. reinhardtii*) | *bbs1-1* | *Lechtreck et al., 2009* | BBS1-null mutant/CC-125 |
| Genetic reagent (*C. reinhardtii*) | BBS3::YFP | This study | | pBKS-gBBS3::YFP-Paro/CC-125 |
| Genetic reagent (*C. reinhardtii*) | BBS3$^{A73L}$::YFP | This study | | pBKS-gBBS3$^{A73L}$::YFP-Paro/CC-125 |
| Genetic reagent (*C. reinhardtii*) | BBS5::YFP | *Xue et al., 2020* | | pBKS-gBBS5::YFP-Paro/CC-125 |
| Genetic reagent (*C. reinhardtii*) | BBS3::YFP$^{5325}$ | This study | | pBKS-gBBS3::YFP-Ble/CC-5325 |
| Genetic reagent (*C. reinhardtii*) | BBS3::YFP$^{clip1}$ | This study | | pBKS-gBBS3::YFP-Ble/*clip1* |
| Genetic reagent (*C. reinhardtii*) | BBS5::YFP$^{5325}$ | *Xue et al., 2020* | | pBKS-gBBS5::YFP-Ble/CC-5325 |
| Genetic reagent (*C. reinhardtii*) | BBS5::YFP$^{clip1}$ | This study | | pBKS-gBBS5::YFP-Ble/*clip1* |
| Genetic reagent (*C. reinhardtii*) | BBS3::YFP$^{1-1}$ | This study | | pBKS-gBBS3::YFP-Ble/*bbs1-1* |
| Genetic reagent (*C. reinhardtii*) | BBS3△N::YFP$^{1-1}$ | This study | | pBKS-gBBS3△N::YFP-Ble/*bbs1-1* |

*Continued on next page*

*Continued*

| Reagent type (species) or resource | Designation | Source or reference | Identifiers | Additional information |
|---|---|---|---|---|
| Genetic reagent (*C. reinhardtii*) | BBS3△N::YFP | This study | | pBKS-gBBS3△N::YFP-Ble/CC-125 |
| Genetic reagent (*C. reinhardtii*) | BBS3$^{miRNA}$ | *Xue et al., 2020* | | pMI-BBS3-Paro/CC-125 |
| Genetic reagent (*C. reinhardtii*) | BBS3$^{Res-WT}$ | *Xue et al., 2020* | | pBKS-gBBS3::GFP-Ble/BBS3$^{miRNA}$ |
| Genetic reagent (*C. reinhardtii*) | BBS3$^{Res-A73L}$ | *Xue et al., 2020* | | pBKS-gBBS3$^{A73L}$::GFP-Ble/BBS3$^{miRNA}$ |
| Genetic reagent (*C. reinhardtii*) | IFT25$^{miRNA}$ | *Dong et al., 2017b* | | pMI-IFT25-Paro/CC-125 |
| Genetic reagent (*C. reinhardtii*) | BBS3$^{Res-△N}$ | This study | | pBKS-gBBS3△N::YFP-Ble/BBS3$^{miNRA}$ |
| Genetic reagent (*C. reinhardtii*) | HR-YFP | This study | | pBKS-HSP 70A-RBCS 2-YFP-Ble/CC-125 |
| Antibody | Anti-IFT46 (rabbit polyclonal) | *Dong et al., 2017b* | | immunoblotting (1:1,000) |
| Antibody | Anti-IFT57 (rabbit polyclonal) | *Dong et al., 2017b* | | immunoblotting (1:1,000) |
| Antibody | Anti-IFT70 (rabbit polyclonal) | *Dong et al., 2017b* | | immunoblotting (1:1,000) |
| Antibody | Anti-BBS1 (rabbit polyclonal) | *Xue et al., 2020* | | immunoblotting (1:1,000) immunofluorescence (1:200) |
| Antibody | Anti-BBS3 (rabbit polyclonal) | *Xue et al., 2020* | | immunoblotting (1:500) immunofluorescence (1:200) |
| Antibody | Anti-BBS4 (rabbit polyclonal) | This study | | immunoblotting (1:500) |
| Antibody | Anti-BBS5 (rabbit polyclonal) | *Xue et al., 2020* | | immunoblotting (1:1000) immunofluorescence (1:200) |
| Antibody | Anti-BBS7 (rabbit polyclonal) | This study | | immunoblotting (1:1000) |
| Antibody | Anti-PLD (rabbit polyclonal) | This study | | immunoblotting (1:1000) |
| Antibody | Anti-α-tubulin (mouse monoclonal) | Sigma-Aldrich | B512 | immunoblotting (1:25,000) |
| Antibody | Anti-Ac-tubulin (mouse monoclonal) | Sigma-Aldrich | 6-11B-1 | immunoblotting (1:50,000) |
| Antibody | Anti-GFP (YFP) (mouse monoclonal) | Roche | 7.1 and 13.1 | immunoblotting (1:1,000) |
| Antibody | HRP-conjugated goat anti-mouse IgG | Jackson Lab. | RRID:AB_10015289 | immunoblotting (1:10,000) |
| Antibody | HRP-conjugated goat anti-rabbit IgG | Jackson Lab. | RRID:AB_2313567 | immunoblotting (1:10,000) |

Continued

| Reagent type (species) or resource | Designation | Source or reference | Identifiers | Additional information |
|---|---|---|---|---|
| Antibody | Alexa-Fluor 594-conjugated goat anti-rabbit IgG | Molecular Probes | RRID:AB_142057 | immunofluorescence (1:400) |
| Antibody | Alexa-Flour 488-conjugated goat anti-mouse IgG | Molecular Probes | RRID:AB_2536097 | immunofluorescence (1:400) |

## Chlamydomonas strains and culture conditions

*Chlamydomonas* strains including CC-125, CC-5325, and the CLiP mutant (LMJ.RY0402.149010) were purchased from the *Chlamydomonas* Genetic Center at the University of Minnesota, Twin Cities, MN (http://www.chlamy.org). BBS1-null mutant *bbs1-1*, BBS3-knockdown strain BBS3$^{miRNA}$, BBS3-rescuing strains BBS3$^{Res-WT}$ and BBS3$^{Res-A73L}$, IFT25-knockdown strain IFT25$^{miRNA}$, BBS5::YFP-expressing strains BBS5::YFP$^{5325}$ and BBS5::YFP have been reported previously (*Dong et al., 2017b*; *Lechtreck et al., 2009*; *Xue et al., 2020*). All strains were listed in the key resources table. Strains were grown in Tris acetic acid phosphate (TAP) medium under continuous light with constant aeration at room temperature. Depending on a specific strain, cells were cultured with or without the addition of 20 µg/ml paromomycin (Sigma-Aldrich), 15 µg/ml bleomycin (Invitrogen), or both antibiotics with 10 µg/ml paromomycin and 5 µg/ml bleomycin.

## Vectors and transgenic strain generation

To express BBS3::YFP, YFP was amplified from pHK214 (*Lv et al., 2017*) by using the primer pair GFP-FOR1 (5'-CGGAATTCATGGCCAAGGGCGAGG-3') and GFP-REV (5'-CCGCTCGAGTTACTTG TACAGCTCGTCCATGC-3') with *Eco*RI and *Xho*I restriction enzyme sites located in its 5'- and 3'-end, respectively, and inserted into *Eco*RI and *Xho*I sites of pBKS-gBBS3::GFP-Ble (*Xue et al., 2020*), resulting in pBKS-gBBS3::YFP-Ble. To express BBS3$^{A73L}$::YFP, the BBS3$^{A73L}$ DNA fragment was cut from pBKS-gBBS3$^{A73L}$::GFP-Paro (*Xue et al., 2020*) and inserted into the *Xba*I and *Eco*RI sites of pBKS-gBBS3::YFP-Ble, resulting in pBKS-gBBS3$^{A73L}$::YFP-Ble. To express BBS3△N::YFP, the BBS3 promoter and BBS3△N fragment was amplified from pBKS-gBBS3::YFP-Ble using the primer pair gBBS3-FOR1 (5'-GCTCTAGATTCCCACTCCCGCAGCC-3') and gBBS3-REV1 (5'-GAACAGCCA-CAACATGCTCTTGTCGCTGTTTGG-3') and gBBS3-FOR2 (5'-CCAAACAGCGACAAGAGCATGTTGC TGGCTGTTC-3') and gBBS3-REV2 (5'-GGAATTCGCTGAGGCGCTCCGCC-3'), respectively. The two fragments were inserted into the *Xba*I and *Eco*RI sites of pBKS-gBBS3::YFP-Ble by three-way ligation, resulting in pBKS-gBBS3△N::YFP-Ble. The BBS5::YFP-expressing vectors pBKS-gBBS5::YFP-Paro and pBKS-gBBS5::YFP-Ble have been described previously (*Xue et al., 2020*). To express YFP, YFP sequence was cut from pBKS-gBBS5::YFP-Ble and inserted into the *Eco*RI and *Xho*I sites of pBKS-HSP70A-RBCS2-GFP-Ble for replacing the GFP segment, resulting in pBKS-HSP70A-RBCS2-YFP-Ble (*Dong et al., 2017a*). After verification by direct nucleotide sequencing, the new constructs were transformed into *Chlamydomonas* strain by electroporation as described previously, and screening of positive transformants was done according to the method described previously (*Xue et al., 2020*).

## DNA and mRNA analysis

*Chlamydomonas* genomic DNA was extracted and purified according to our protocol reported previously (*Dong et al., 2017b*). To amplify BBS3 genomic sequence, 20 ng of genomic DNA was used as template and the PCR reaction was performed at 95℃ for 2 min followed by 30 cycles of 95℃ for 10 s, 61℃ for 30 s, and 72℃ for 4 min with the primer pair gBBS3-FOR (5'-GCTCTAGATTCCCAC TCCCGCAGCC-3') and gBBS3-REV (5'-GGAATTCGCTGAGGCGCTCCGCC-3'). For BBS3 mRNA analysis, total RNA was extracted according to our previously reported protocol (*Dong et al., 2017b*). Five micrograms of mRNA was reverse-transcribed at 42℃ for 1 hr with M-MLV Reverse Transcriptase (Promega) and 3'-RACE Oligo (dT) primer (5'-GCCTAGCGTCGTTCCAGCAGTGA TTTACGCGTCGACTAGTACTTTTTTTTTTTTTTTTTTT-3'). PCR was performed to amplify BBS3 cDNA

with primer pair cBBS3-FOR (5'-CCGGATCCATGGGCTTCTTTGACAAG-3') and cBBS3-REV (5'-CCGAATTCGCTGAGGCGCTCCGCCAG-3'). 3'-RACE-PCR was performed to determine the exact 3'-end nucleotide sequence of BBS3 cDNA with the primer pair cBBS3-FOR1 (5'-CCGAATTC TGGCAATGCCACCCGTGGAAATCGCC-3') and cBBS3-3'-RACE-REV (5'-CCCTCGAGTAGCGTCG TTCCAGCAGTGATTTAC-3'). PCRs were performed at 95℃ for 30 s followed by 30 cycles of 95℃ for 30 s, 57℃ for 30 s, and 72℃ for 1 min. The amplified PCR products were then visualized in agarose gel and purified for nucleotide sequencing.

## Antibodies and immunoblotting

Polyclonal antibodies raised in rabbits have been described previously (*Dong et al., 2017b*; *Xue et al., 2020*; *Zhu et al., 2017*). Rabbit-originated antibodies against BBS4, BBS7, and PLD were generated by Beijing Protein Innovation. Monoclonal antibodies against GFP (YFP) (mAbs 7.1 and 13.1, Roche), α-tubulin (mAb B512, Sigma), and Ac-tubulin (mAb 6-11B-1, Sigma-Aldrich) were commercially purchased (key resources table). Preparation of whole cell, cell body, and ciliary extracts, SDS-PAGE electrophoresis, and immunoblotting were done as detailed previously (*Xue et al., 2020*). If not otherwise specified, 20 μg of total protein from each sample was loaded for SDS-PAGE. In immunoblotting assays, dilution used for primary and secondary antibodies is listed in the key resources table.

## Isolation of cilia and cell bodies

Methods for the isolation of cilia and cell bodies have been described previously (*Xue et al., 2020*). Briefly, cells were suspended in 150 mL of TAP (pH 7.4) and incubated for 2~4 hr under strong light with bubbling. Then, 0.5 M acetic acid was added to reach a pH value of 4.5 to deciliate cells before 0.5 M KOH was added to reach a pH value of 7.4. After centrifugation (600 $\times g$ at 4℃ for 5 min), cell bodies (pellets) and cilia (supernatants) were collected separately. Cilia were then repeatedly washed with HMDEK (10 mM Hepes, pH 7.4, 5 mM $MgSO_4$, 1 mM dithiothreitol [DTT], 5 mM EDTA, and 25 mM KCl) by centrifugation (14,000 $\times g$ at 4℃ for 10 min) until the green color disappeared completely.

## Preparation of ciliary fractions

After cilia were isolated, they were dissolved in HMDEK buffer (50 mM HEPES pH 7.2, 5 mM $MgCl_2$, 1 mM DTT, 0.5 mM EDTA, and 25 mM KCl) plus protein inhibitors (PI) (1 mM PMSF, 50 μg/ml soybean trypsin inhibitor, 1 μg/ml pepstatin A, 2 μg/ml aprotinin, and 1 μg/ml leupeptin) and fresh-frozen in liquid nitrogen. The ciliary matrix fraction was obtained by freezing and thawing cilia followed by centrifugation (27,000 $\times g$, 4℃, 15 min), and the pellets were dissolved in an HMEDK buffer containing 0.5% NP-40 and stayed on ice for 15 min. The supernatant and pellet were collected after centrifugation (14,000 $\times g$ at 4℃ for 10 min) as membrane and axonemal fractions, respectively, according to a previously described method (*Liang et al., 2014*).

## Preparation of liposome and binding assays

Liposomes were prepared exactly as described previously (*Jin et al., 2010*). BBS3 association with liposome was conducted in an HMEK buffer. Liposomes (2 μg) were incubated with 100 μg of bacterially expressed C-terminal 6×His tagged BBS3::YFP or BBS3△N::YFP in the presence of GTPγS (100 mM), GDP (100 mM), or neither in a 100 μL reaction at 30℃ for 1 hr. The reactions were centrifuged at 385,000 $\times g$ for 30 min at 24℃, and equal portions of the resulting pellets were resolved by SDS-PAGE and immunoblotted with α-BBS3.

## Immunoprecipitation

Cell bodies and cilia isolated from *Chlamydomonas* strains expressing YFP-tagged BBS3 or its variants were resuspended in HMEK+PI supplemented with 50 mM NaCl and lysed by adding nonidet P-40 (NP-40) to 1%. The supernatants were collected by centrifugation (14,000 $\times g$, 4℃, 10 min) and incubated with agitation with 5% BSA-pretreated camel anti-GFP antibody-conjugated agarose beads (V-nanoab Biotechnology) for 2 hr at 4℃. The beads were then washed with HMEK containing 150 mM NaCl, 50 mM NaCl, and finally 0 mM NaCl. The beads were then added with Laemmli SDS

sample buffer and boiled for 5 min before centrifuging at 2,500 ×*g* for 5 min. The supernatants were then analyzed by immunoblotting as described above.

## Immunofluorescence

Immunofluorescence staining was performed according to our published method (*Fan et al., 2010*). The primary antibodies against PLD, YFP, BBS3, BBS1, and BBS5 and the secondary antibodies – Alexa-Fluor594-conjugated goat anti-rabbit IgG and Alexa-Fluor488-conjugated goat anti-mouse IgG (Molecular Probes, Eugene, OR) – are listed in the key resources table with their suggested dilutions for immunofluorescence staining. Images were captured with an IX83 inverted fluorescent microscope (Olympus) equipped with a back illuminated scientific CMOS camera (Prime 95B, Photometrics) at 100× amplification and processed with CellSens Dimension (version 2.1, Olympus).

## Sucrose density gradient centrifugation

Sucrose density gradient centrifugation of ciliary extracts was performed according to our published method (*Wang et al., 2009*). In brief, linear 12 mL of 10–25% sucrose density gradients in 1× HMDEK buffer plus protease inhibitors and 1% NP-40 were generated using the Jule Gradient Former (Jule Inc, Milford, CT) and used within 1 hr. The cilia were opened with liquid nitrogen for three rounds of frozen-and-thaw cycles and centrifuged at 12,000 rpm, 4°C, for 10 min. Seven hundred microliters of ciliary extracts were then loaded on the top of the gradients and separated at 38,000 rpm, 4°C, for 14 hr in a SW41Ti rotor (Beckman Coulter). The gradients were fractioned into 24–26 0.5 ml aliquots using a Pharmacia LKB Pump P-1 coupled with a FRAC-100 fraction collector. The standards used to calculate S-values were BSA (4.4S), aldolase (7.35S), catalase (11.3S), and thyroglobulin (19.4S). Twenty microliters of each fraction was analyzed by immunoblotting as described above. If necessary, the assay was carried out in the presence of GTPγS (20 mM) or GDP (20 mM), respectively.

## IFT video imaging and speed measurements

The motility of GFP- and YFP-tagged proteins in cilia was imaged at ~15 fps using TIRF microscopy on an inverted microscope (IX83, Olympus) equipped with a through-the-objective TIRF system, a 100×/1.49 NA TIRF oil immersion objective (Olympus), and a back-illuminated scientific CMOS camera (Prime 95B, Photometrics) as detailed previously (*Xue et al., 2020*). To quantify IFT speeds and frequencies, kymograms were generated and measured with CellSens Dimension (version 2.1, Olympus).

## Protein expression and purification

The cDNAs encoding BBS3, BBS3$^{A73L}$, and BBS3$^{T31R}$ were amplified from pGEX-6P-cBBS3, pGEX-6P-cBBS3$^{A73L}$, and pGEX-6P-cBBS3$^{T31R}$ using the primer pair cBBS3-FOR and cBBS3-REV as described earlier. The cDNAs of BBS3 and its mutants were inserted into the *Bam*HI and *Hin*d III sites of pET-28a (Novagen) to result in pET-28a-cBBS3, pET-28a-cBBS3$^{A73L}$, and pET-28a-cBBS3$^{T31R}$, respectively. To express YFP, the cDNA of YFP was cut from pBKS-gBBS3::YFP-Ble with *Eco*RI and *Xho*I and inserted into pET-28a (Novagen) to result in pET-28a-YFP. To express the C-terminal YFP-tagged BBS3 and BBS3ΔN, the cDNAs of BBS3 and BBS3ΔN were amplified from pGEX-6P-cBBS3 using the primer pair cBBS3-FOR and cBBS3-REV (for BBS3) and cBBS3ΔN-FOR (5′-CCGGATCCATGCTCTTGTCGCTG-3′) and cBBS3-REV (for BBS3ΔN) and inserted into *Bam*HI and *Eco*RI sites of pET-28a-YFP to result in pET-28a-BBS3::YFP and pET-28a-BBS3ΔN::YFP. After these plasmids were transformed into bacteria, the bacterially expressed recombinant proteins were purified with Ni-NTA beads and cleaved with thrombin (Solarbio) to get rid of the N-terminal 6× His tag according to our previous report (*Xue et al., 2020*). If necessary, 10 μg of proteins from elutes was resolved on 12% SDS-PAGE gels and visualized with Coomassie Blue staining.

## Small GTPase assay

Small GTPase assay was performed according to our previous report (*Xue et al., 2020*). Intrinsic GTP hydrolysis of the bacterially expressed BBS3, BBS3$^{A73L}$, and BBS3$^{T31R}$ was measured by optical assay for the release of inorganic phosphate with reagents from the QuantiChrom ATPase/GTPase assay kit (Bioassay Systems) (*Pan et al., 2006*).

## Statistical analysis

Statistical analysis was performed with GraphPad Prism 8.0 (GraphPad Software). For comparisons on speeds and frequencies of the GFP- and YFP-labeled proteins, one-sample unpaired Student's $t$-test was used. Data were presented as mean ± SD from three independently performed experiments, and n means sample numbers. n.s. represents non-significance. If not mentioned elsewhere, significance was set as $p < 0.05$.

## Acknowledgements

This work was supported by National Natural Science Foundation of China grant 32070698, Tianjin Municipal Science and Technology Bureau grants 19PTSYJC00050 and 18JCZDJC34100, and International Center for Genetic Engineering and Biotechnology grant CRP/CHN15-01 (ZCF), National Natural Science Foundation of China grant 41876134 (JS), and the National Institutes of Health grant GM110413 (KFL). The content is solely the responsibility of the authors and does not necessarily represent the official views of the National Institutes of Health.

## Additional information

### Funding

| Funder | Grant reference number | Author |
| --- | --- | --- |
| National Natural Science Foundation of China | 32070698 | Zhen-Chuan Fan |
| Tianjin Municipal Science and Technology Commission | 18JCZDJC34100 | Zhen-Chuan Fan |
| Tianjin Municipal Science and Technology Commission | 19PTSYJC00050 | Zhen-Chuan Fan |
| International Center for Genetic Engineering and Biotechnology | CRP/CHN15-01 | Zhen-Chuan Fan |
| National Natural Science Foundation of China | 41876134 | Jun Sun |
| National Institutes of Health | GM110413 | Karl F Lechtreck |

The funders had no role in study design, data collection and interpretation, or the decision to submit the work for publication.

### Author contributions

Yan-Xia Liu, Data curation, Formal analysis, Validation, Investigation, Visualization, Methodology, Project administration; Bin Xue, Data curation, Formal analysis, Validation, Investigation, Visualization, Methodology; Wei-Yue Sun, Investigation, Methodology; Jenna L Wingfield, Investigation, Writing - review and editing; Jun Sun, Resources, Funding acquisition; Mingfu Wu, Zhenlong Wu, Formal analysis; Karl F Lechtreck, Funding acquisition, Investigation; Zhen-Chuan Fan, Conceptualization, Resources, Data curation, Supervision, Funding acquisition, Validation, Writing - original draft, Project administration, Writing - review and editing

### Author ORCIDs

Zhen-Chuan Fan https://orcid.org/0000-0003-1712-3413

### Decision letter and Author response

Decision letter https://doi.org/10.7554/eLife.59119.sa1
Author response https://doi.org/10.7554/eLife.59119.sa2

## Additional files

### Supplementary files
• Transparent reporting form

### Data availability
All data generated or analysed during this study are included in the manuscript and supporting files.

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
