## [Decision Letter]

**Acceptance summary:**

This manuscript is of broad interest to scientists in the field of ciliary biology. The identification of BBS3 for entry into cilia separately from BBsome and its role in mediating membrane protein exit from cilia have deepened our understanding of the mechanism of ciliary exit of membrane proteins.

**Decision letter after peer review:**

Thank you for submitting your article "Bardet-Biedl Syndrome 3 Protein Mediates Phototaxis through Promoting Ciliary Exit of Phospholipase D via the BBSome" for consideration by *eLife*. Your article has been reviewed by three peer reviewers, one of whom is a member of our Board of Reviewing Editors, and the evaluation has been overseen by Anna Akhmanova as the Senior Editor. The reviewers have opted to remain anonymous.

The reviewers have discussed the reviews with one another and the Reviewing Editor has drafted this decision to help you prepare a revised submission.

Summary:

BBSome is critical for membrane protein trafficking within cilia. The current work demonstrates that BBS3 enters and exits cilia separately from BBsome, and functions at the ciliary tip to load membrane proteins for their exit, which are unique findings. However, several major concerns were raised about the data to support the model proposed in the manuscript. We would like the authors to consider the points listed below should a revision be prepared.

Essential revisions:

1) This manuscript proposed that GTP form is required for ciliary entry of BBS3 and for loading PLD at the ciliary tip. However, no direct in vivo data were shown to support this. Expressing a GTP form in the BBS3 RNAi strain or other BBS3 null mutants, is needed to show how the GTP form behaves (such as, where it localizes in the cilia, how it moves along the cilia) in cilia and if PLD accumulates at the ciliary tip.

2) The claim that ciliary entry of BBS3 does not require BBSome was demonstrated in a *bbs-1* null mutant. However, in Figure 3C, it showed that BBS1 is still present in the cell extracts of *bbs-1* mutant. Thus this claim is not supported by the data though in Figure 2A it showed BBS1 was not present in the mutant. New experiments should be performed to clarify this.

3) Some critical data are derived from using the *clip1* mutant. However, the authors claimed that the *clip1* strain must contain other unidentified mutation(s) because an apparently wild-type BBS3 protein is expressed. Since the nature of this "other unidentified mutation(s)" is unclear, it raises serious concerns about the conclusions made using the *clip1* mutant. Therefore, it is requested to look more carefully at the BBS3 mutation in the *clip1* mutant. Given the limited information the authors supply in the manuscript, it is still possible that the BBS3 protein detected in the *clip1* mutant is slightly truncated/modified at the C-terminus. If BBS3 expressed in *clip1* is modified at the C-terminus, this indicates that the C-terminus in addition to its N-terminus is critical for its transport into flagella and lack of C-terminus is dominant negative. The experiments needed is to look at the full-length BBS3 cDNA (a 3' end RACE might suffice). If cDNA sequence shows that a modified BBS3 protein is expressed in the *clip1* mutant, we request to confirm that the BBSome phenotype in this *clip1* mutant is caused by this mutation by expressing this form of BBS3 protein in a BBS3 RNAi strain or a BBS3 null strain. And the conclusions obtained using the *clip1* mutant in this manuscript is legitimate. If cDNA sequence shows that a wild-type BBS3 protein is expressed in *clip1* mutant, we request to confirm any conclusions obtained using the *clip1* mutant with a cell line that has the same phenotype but the mutation background is clear (a good candidate would be a BBS3RNAi strain that expresses BBS3ΔN).

4) BBS3ΔN expressed to a level barely exceeding that of the remaining endogenous BBS3 (Figure 7D) was able to rescue the flagellar levels of BBSome components to an extent that appears to exceed those in the wild-type flagella (Figure 7E). It could mean that BBSome was accumulated in the cilia due to lack of retrograde transport, thus arguing against the claim of a BBS3-independent IFT of BBSome (Figure 6H) based solely on the weird *clip1* strain. Therefore, a critical experiment would be to examine flagellar BBSome trafficking in this strain or preferably in a rescue strain expressing a higher level (e.g. at least similar to the wild-type level) of BBS3ΔN.

5) Figure 1F: IP lacks negative control. Lysate from a strain expressing YFP alone or an irrelevant YFP-fusion protein should be used as negative control.

6) Figure 3B: I suggest the authors show either the input or the quality of the bacterially-expressed proteins.

7) Demonstrate a BBS3-GTP-dependent association of PLD with BBSome in fractionation experiments like those in Figure 5A and C.

8) An interesting finding of this paper is that BBS3 doesn't undergo IFT within cilia. However, this conclusion was drawn based on the TIRF results using cell lines with wild-type BBS3 background. It is requested to check tagged BBS3 movement in a BBS3-null background or in a BBS3 RNAi strain (The BBS3^res-WT^ strain in Xue et al., 2020, might be a good candidate).

---

## [Author Response]

Essential revisions:1) This manuscript proposed that GTP form is required for ciliary entry of BBS3 and for loading PLD at the ciliary tip. However, no direct in vivo data were shown to support this. Expressing a GTP form in the BBS3 RNAi strain or other BBS3 null mutants, is needed to show how the GTP form behaves (such as, where it localizes in the cilia, how it moves along the cilia) in cilia and if PLD accumulates at the ciliary tip.

Our previous study has shown that GTP- but not GDP-bound BBS3 enters cilia (Xue et al., 2020). In the current study, we actually have provided in vivo evidence by using both the *clip1* mutant and the BBS3-knockdown strain BBS3^miRNA^ to show that BBS3 promotes PLD loading onto the BBSome at the ciliary tip for ciliary exit (Figure 7). In addition, we have performed ciliary fraction assays to show that the GTP-bound mutant BBS3^A73L^::YFP localizes in the ciliary membrane (Figure 4E and F). In our revised manuscript, we have performed TIRF analysis, as requested, to determine movement of BBS3^A73L^::YFP in the BBS3-knockdown strain BBS3^miRNA^ (Figure 4D). Similar to BBS3^A73L^::YFP expressed in wild-type CC-125 cells (Figure 4A-C), BBS3^A73L^::YFP undergoes diffusion in cilia of BBS3^miRNA^ cells (Figure 4D). We also have found that BBS3^A73L^::YFP restored the accumulated PLD to wild-type level in cilia of BBS3^miRNA^ cells, revealing that GTPase cycling of BBS3 is not required for PLD turnaround at the ciliary tip for ciliary exit (Figure 7D and E). Please see our revised manuscript for details.

2) The claim that ciliary entry of BBS3 does not require BBSome was demonstrated in a bbs-1 null mutant. However, in Figure 3C, it showed that BBS1 is still present in the cell extracts of bbs-1 mutant. Thus this claim is not supported by the data though in Figure 2A it showed BBS1 was not present in the mutant. New experiments should be performed to clarify this.

We have accidently combined a wrong band of BBS1 into Figure 3C. In the revised manuscript, we have performed western blotting assay again to assure that the *bbs1-1* mutant is indeed a BBS1-null mutant and have integrated the correct band into Figure 3C. Please see our revised Figure 3C for details.

3) Some critical data are derived from using the clip1 mutant. However, the authors claimed that the clip1 strain must contain other unidentified mutation(s) because an apparently wild-type BBS3 protein is expressed. Since the nature of this "other unidentified mutation(s)" is unclear, it raises serious concerns about the conclusions made using the clip1 mutant. Therefore, it is requested to look more carefully at the BBS3 mutation in the clip1 mutant. Given the limited information the authors supply in the manuscript, it is still possible that the BBS3 protein detected in the clip1 mutant is slightly truncated/modified at the C-terminus. If BBS3 expressed in clip1 is modified at the C-terminus, this indicates that the C-terminus in addition to its N-terminus is critical for its transport into flagella and lack of C-terminus is dominant negative. The experiments needed is to look at the full-length BBS3 cDNA (a 3' end RACE might suffice). If cDNA sequence shows that a modified BBS3 protein is expressed in the clip1 mutant, we request to confirm that the BBSome phenotype in this clip1 mutant is caused by this mutation by expressing this form of BBS3 protein in a BBS3 RNAi strain or a BBS3 null strain. And the conclusions obtained using the clip1 mutant in this manuscript is legitimate. If cDNA sequence shows that a wild-type BBS3 protein is expressed in clip1 mutant, we request to confirm any conclusions obtained using the clip1 mutant with a cell line that has the same phenotype but the mutation background is clear (a good candidate would be a BBS3RNAi strain that expresses BBS3ΔN).

Similar to the observation that the endogenous BBS3 only enriches at the basal bodies but is not able to enter cilia in *clip1* mutant cells (Figure 1B and E), a C-terminal YFP-tagged BBS3 (BBS3::YFP) also targets to the basal bodies but cannot enter cilia in BBS3::YFP^clip1^ cells, confirming that wild-type BBS3 itself loses its ability for ciliary entry in the *clip1* mutant (Figure 1D). As requested, we performed 3*'*-end RACE analysis to determine the exact 3*'*-end sequence of BBS3 mRNA in *clip1* mutant and found that *clip1* mutant contains wild-type BBS3 mRNA, revealing that BBS3 itself remains intact in *clip1* mutant but loses its ability to enter cilia (please see Figure 1—figure supplement 1F and G). In our original manuscript, we have confirmed this observation by expressing BBS3△N::YFP in the BBS3-knockdown strain BBS3^miRNA^ (Figure 7D and E). Our data showed that BBS3△N::YFP binds and targets the BBSome to the basal bodies but does not enter cilia itself. By such a way, BBS3△N::YFP rescues the BBSome to wild-type level in cilia, confirming that the absence of BBS3 in cilia does not affect ciliary content of the BBSome. In the revised section entitled “BBS3 is not required for the BBSome to enter cilia from the basal body”, we accordingly deleted the sentence “we propose that the *clip1* strain must contain other unidentified mutation(s)” and replaced “To verify this observation” with “To verify that BBS3 is not able to enter cilia in *clip1* cells” and added one sentence of “This notion was confirmed when BBS3 knockdown and rescue studies were performed (Figure 7D and E)”.

In the revised section entitled “BBS3 is essential for PLD to associate with the BBSome for ciliary exit”, we emphasized this by rewriting sentences as “When BBS3::GFP was expressed in BBS3^miRNA^ cells (resulting the strain BBS3^Res-WT^) it entered cilia and restored BBS1, BBS5 and PLD to wild-type levels (Figure 7D-F) (Xue et al., 2020). […] In addition, BBS3△N::YFP did not enter cilia itself but restored BBS1 and BBS5 to wild-type levels in cilia of the BBS3-knockdown BBS3^miRNA^ cells (Figure 7D-F), providing compelling evidence to confirm that ciliary entry of the BBSome from the ciliary base does not depend on BBS3 (Figure 1).”

4) BBS3ΔN expressed to a level barely exceeding that of the remaining endogenous BBS3 (Figure 7D) was able to rescue the flagellar levels of BBSome components to an extent that appears to exceed those in the wild-type flagella (Figure 7E). It could mean that BBSome was accumulated in the cilia due to lack of retrograde transport, thus arguing against the claim of a BBS3-independent IFT of BBSome (Figure 6H) based solely on the weird clip1 strain. Therefore, a critical experiment would be to examine flagellar BBSome trafficking in this strain or preferably in a rescue strain expressing a higher level (e.g. at least similar to the wild-type level) of BBS3ΔN.

As requested, we have carefully conducted the assay again and found that the selected rescue strain expresses BBS3△N::YFP and endogenous BBS3 to a combined amount equivalent to BBS3 alone in CC-125 cells. Similar to the *clip1* mutant in which ciliary content of the BBSome is not affected by BBS3, BBS3△N::YFP rescues the BBSome components to wild-type levels in cilia of the BBS3-knockdown BBS3^miRNA^ cells, revealing that ciliary presence of BBS3 is not required for maintaining BBSome content at wild-type level in cilia (Figure 7D and E). Please see our revised Figure 7D and E for details.

5) Figure 1F: IP lacks negative control. Lysate from a strain expressing YFP alone or an irrelevant YFP-fusion protein should be used as negative control.

As requested, we have expressed YFP alone in CC-125 cells and added lysate from this YFP-expressing strain as a negative control in Figure 1F in our revised manuscript. Please see our revised Figure 1F for details.

6) Figure 3B: I suggest the authors show either the input or the quality of the bacterially-expressed proteins.

As suggested, we have added the Coomassie staining of SDS-PAGE of the purified BBS3::YFP and BBS3△N::YFP to show the input and quality of the bacterially-expressed proteins in Figure 3B. Please see our revised Figure 3B for details.

7) Demonstrate a BBS3-GTP-dependent association of PLD with BBSome in fractionation experiments like those in Figure 5A and C.

As requested, we have performed immunoblotting assays to determine the BBS3-GTP-dependent association of PLD with the BBSome in sucrose density gradient centrifugation. As shown in Figure 7H, PLD rarely co-sediments with the BBSome components BBS1 and BBS5 in ciliary extracts of *clip1* cells but primarily co-fractions with the BBSome in ciliary extracts of BBS3^Res-A73L^ but not BBS3^Res-△N^ cells. We emphasized this in the revised section entitled “BBS3 is essential for PLD to associate with the BBSome for ciliary exit” by adding sentences as “Together with the observation that PLD rarely co-sedimented with the BBSome (checked with BBS1 and BBS5) in ciliary extracts of *clip1* cells; a minority of PLD co-sedimented with the BBSome in ciliary extracts of BBS3^Res-△N^ cells; and the majority of PLD became co-sedimented with the BBSome in ciliary extracts of BBS3^Res-A73L^ cells (Figure 7H), our data suggest that GTP-bound BBS3 efficiently enables PLD to associate with the BBSome in cilia at the ciliary tip to undergo retrograde IFT (Figure 7I).”

8) An interesting finding of this paper is that BBS3 doesn't undergo IFT within cilia. However, this conclusion was drawn based on the TIRF results using cell lines with wild-type BBS3 background. It is requested to check tagged BBS3 movement in a BBS3-null background or in a BBS3 RNAi strain (The BBS3^res-WT^ strain in Xue et al., 2020, might be a good candidate).

As requested, we have performed TIRF analysis on BBS3^Res-WT^ and BBS3^Res-A73L^ cells (please see Figure 4D; and Figure 4—video3 and 4). BBS3::GFP and BBS3^A73L^::GFP both showed a pattern of diffusion but not IFT in cilia of BBS3^Res-WT^ and BBS3^Res-A73L^ cells. We described this in the section entitled “BBS3 associates with the ciliary membrane in a GTP-dependent manner” as “Similar stationary pattern was also recorded for the C-terminal green fluorescent protein (GFP)-tagged BBS3 and the A73L mutant in the rescuing strains BBS3^Res-WT^ and BBS3^Res-A73L^, in which BBS3::GFP and BBS3^A73L^::GFP are expressed in similar amounts in the BBS3-knockdown strain BBS3^miRNA^ (Figure 4D; and Figure 4—videos 3 and 4) (Xue et al., 2020)”. Please see our revised manuscript for details.